# LOTTERY: Learning from Reference-Only Samples in Two-Sample Testing under Size Asymmetry

Xunye Tian [* 1 2]   Zhijian Zhou [* 1]   Liuhua Peng [1]   Feng Liu [1]

## Abstract

Data-adaptive two-sample testing assesses if two samples come from the same distribution, using a discrepancy learned from the data (e.g., via kernel-based feature representations). Such methods typically rely on data splitting to decouple learning from testing and control type I error. However, this paradigm is ill-suited to few-shot settings with severe sample-size imbalance: abundant reference samples are available, while only a handful of query samples arrive. In this paper, we show how this imbalance can be leveraged constructively. Using abundant reference data, we learn reference-dependent representations that summarize salient structure of the reference distribution and provide informative *signals* for detecting departures. We incorporate a collection of representation families that capture both global and local structure, and adaptively weight them using only reference samples via an uncertainty-guided principle. Theoretically, we establish permutation-based type I error control and show consistency of the aggregated test: as the sample sizes grow, the test power converges to one whenever the representation set contains at least one consistent representation. Empirically, our aggregation achieves strong performance across a range of benchmarks while retaining type I error control. The code of our LOTTERY is available at `github.com/yeager20001118/lottery-two-sample-testing-paper`

## 1. Introduction

Two-sample testing is a core primitive in modern machine learning, underpinning a wide range of real-world applications where one must determine whether newly observed

[*]Equal contribution  [1]University of Melbourne [2]Maincode. Correspondence to: Feng Liu <fengliu.ml@gmail.com>.

*Proceedings of the 43rd International Conference on Machine Learning*, Seoul, South Korea. PMLR 306, 2026. Copyright 2026 by the author(s).

data follow the same distribution. Prominent examples include dataset shift detection and model monitoring (Rabanser et al., 2019), adversarial and out-of-distribution detection in vision systems (Grosse et al., 2017; Liu et al., 2025), machine-generated text detection and content authenticity verification (Zhang et al., 2024), as well as membership-inference attack detection (Li et al., 2021). More broadly, two-sample testing is a foundational tool for trustworthy ML, enabling practitioners to audit model behaviour, detect distributional violations, and guard against silent failures arising from distribution drift, privacy leakage, or malicious manipulation (Szegedy et al., 2014; Han et al., 2025). In many real-world pipelines, users maintain a large historical reference dataset presumed to reflect nominal operation, and must repeatedly assess whether a newly arriving batch of query samples continues to follow the same distribution.

**Related works.** Most nonparametric two-sample tests are based on kernel methods, where *maximum mean discrepancy* MMD compares kernel mean embeddings of the two samples with a fixed kernel bandwidth (Berlinet & Thomas-Agnan, 2004; Gretton et al., 2012). Modern variants often improve test power by *aggregating* MMD statistics over multiple kernel bandwidths (e.g., MMDAgg or MMD-FUSE), yielding strong power across diverse alternatives while maintaining type I error control (Schrab et al., 2023; Biggs et al., 2023). While these kernel-based tests can be effective with well-chosen kernels, selecting a representation that is sensitive to the unknown alternative becomes challenging in complex, high-dimensional settings. This motivates *learning-based* two-sample testing, where one adapts features, kernels, or representations directly from data to improve power, while maintaining valid type I error control via resampling-based calibration (Lopez-Paz & Oquab, 2017; Sutherland et al., 2017; Liu et al., 2020). A common thread across these methods is an explicit *train–test split* to ensure valid type I error when the discrepancy is adapted to the data.

**The size-asymmetry and few-shot bottleneck.** Despite their empirical success in balanced-sample settings, existing learning-based tests are ill-suited to a ubiquitous monitoring scenario where query data arrive in *extremely small batches* (Gao et al., 2021; Li et al., 2021). In practice, one may accumulate abundant reference data over time, yet only

observe a handful of query points per test, due to latency constraints, rare-event settings, or limited observation windows. In this scenario, a train–test split becomes particularly damaging: using even a few query points for learning yields unstable discrepancies, while leaving too few for testing produces noisy calibration and low power. Crucially, these few-shot limitations cannot be mitigated simply by collecting more reference data, since the fundamental bottleneck is the scarcity of query points. Moreover, the information contained in the abundant reference set is often underutilized by standard learning-based pipelines, since adaptivity is typically driven by learning from the pooled batch rather than leveraging the full reference sample (Liu et al., 2020).

This paper asks: *how can we retain the adaptivity of learning-based testing while avoiding learning from scarce query data?* Our key observation is that, under size asymmetry and few-shot, it is often more natural to treat $Y$ as a *query* for testing only rather than as a dataset for training.

**A reference-only viewpoint.** We propose to learn from the reference distribution alone: using abundant reference data, we construct *reference-dependent representations* (RDR) that summarize salient global and local structure of $\mathbb{P}$ and output a scalar compatibility score for any input. Testing then reduces to checking whether the query batch induces significantly different scores than reference data. This one-sided perspective is inspired by, and empirically supported by, the success of one-class classification and anomaly detection: large amount of works show that learning features from a single (in-distribution) class can yield representations in which out-of-distribution points become easier to separate and detect (Tax & Duin, 2004; Ruff et al., 2018; Goyal et al., 2020; Liznerski et al., 2021; Zong et al., 2018; Sehwag et al., 2021; Ruff et al., 2021; Chen et al., 2022).

However, a single reference-dependent representation cannot be uniformly powerful against heterogeneous alternatives. To address this, we construct a *family* of complementary RDRs that probe multiple aspects of $\mathbb{P}$, including both global and local structure. We then aggregate their evidence while maintaining nonparametric validity.

**LOTTERY.** We introduce **LOTTERY** (*Learning from reference-Only samples in Two-sample Testing under sizE asymmetRY*), a framework that (i) learns multiple complementary RDRs using only reference data, (ii) aggregate and weighted select them into a single test statistic, and (iii) performs a pooled permutation test that yields exact finite-sample type I error control. At a high level, LOTTERY replaces the usual "learn a discrepancy between $X$ and $Y$" paradigm with "learn a collection of compatibility signals from $X$ and test $Y$ against them".

**Contributions.** We make three main contributions. First, we introduce a reference-only framework for two-sample

testing in the *highly size-asymmetric, few-shot regime*, where abundant reference samples are available but only small query batches arrive at test time. Second, we propose an uncertainty-guided aggregation procedure that adaptively weights reference-dependent representations, *prioritizing stable and informative signals* for detecting departures from the reference distribution. Third, we provide finite-sample type I error control and establish consistency of test power, and empirically demonstrate strong performance across synthetic benchmarks and real-world application datasets.

**Overview.** Section 2 reviews asymmetric two-sample testing and the limitations of split-based data-adaptive tests. Section 3 highlights the size-asymmetry bottleneck and motivates reference-only learning. Section 4 introduces LOTTERY, including its RDR families. Section 5 presents our uncertainty-guided weighting/selection mechanism. Section 6 provides theoretical guarantees. Section 7 reports experimental results. Appendix A and B contain theoretical proofs and experimental details.

## 2. Preliminaries

**Asymmetric Two-sample Testing.** Given two unknown distributions $\mathbb{P}$ and $\mathbb{Q}$ on $(\mathcal{X}, \mathscr{A}(\mathcal{X}))$, we observe a reference sample $X = \{\boldsymbol{x}_1, \dots, \boldsymbol{x}_n\} \overset{\text{i.i.d.}}{\sim} \mathbb{P}$ and a query sample $Y = \{\boldsymbol{y}_1, \dots, \boldsymbol{y}_m\} \overset{\text{i.i.d.}}{\sim} \mathbb{Q}$. We focus on the *sample-size asymmetric* regime, where the reference distribution is well sampled (large $n$) but the query batch is limited (small $m$), as commonly arises in monitoring and detection applications. The goal is to test

$$H_0 : \mathbb{P} = \mathbb{Q} \quad \text{vs.} \quad H_1 : \mathbb{P} \neq \mathbb{Q}.$$

A two-sample test is a measurable function $\phi : \mathcal{X}^n \times \mathcal{X}^m \to \{0, 1\}$, where $\phi(X, Y) = 1$ denotes *rejecting $H_0$* and $\phi(X, Y) = 0$ denotes *not rejecting $H_0$*. The test controls the type I error at level $\alpha \in (0, 1)$ if, under the null hypothesis,

$$\mathbb{P}\big(\phi(X, Y) = 1\big) \leq \alpha.$$

For a fixed alternative distribution pair $(\mathbb{P}, \mathbb{Q})$ with $\mathbb{P} \neq \mathbb{Q}$, the (finite-sample) power of $\phi$ is

$$\mathbb{P}\big(\phi(X, Y) = 1\big),$$

i.e., the probability of correctly rejecting $H_0$ under the alternative hypothesis $H_1$.

**Data-Adaptive Testing with Train-Test Splitting.** A standard paradigm in data-adaptive two-sample testing is to decouple representation learning from hypothesis testing via an explicit train-test split. Given the pooled sample $Z = X \cup Y$, one constructs disjoint subsets $Z^{\text{tr}} = X^{\text{tr}} \cup Y^{\text{tr}}$ and $Z^{\text{te}} = X^{\text{te}} \cup Y^{\text{te}}$. A representation mapping $f$ is learned on $Z^{\text{tr}}$ to increase a data-driven measure of discrepancy between the two samples (e.g., by maximizing a

chosen objective or training a classifier to separate them) (Lopez-Paz & Oquab, 2017; Liu et al., 2020; Kübler et al., 2022; Tian et al., 2025). The learned mapping is then evaluated on $Z^{\text{te}}$ to construct a test statistic, whose significance is assessed by permutation testing. This train-test separation is crucial for valid type I error control, as it prevents using the same data both to adaptively choose the representation and to evaluate its evidence against the null.

## 3. Motivation

In this section, we introduce the main challenges of current data-adaptive two-sample testing framework under the imbalanced size case of two samples.

### 3.1. Limitations of Train-test Splitting

Building on the train–test split paradigm described in Section 2, we show that this design becomes problematic in the severely imbalanced two-sample regime.

When $m = |Y|$ is limited, further splitting the query sample yields extremely small $Y^{\text{tr}}$ and $Y^{\text{te}}$, so the learned representations are often dominated by sampling noise. Learning from such small training subsets leads to high-variance and unstable feature extractors, a phenomenon consistent with classical generalization bounds and empirical observations in low-sample regimes (Shalev-Shwartz & Ben-David, 2014; Zhang et al., 2017). Moreover, allocating only a handful of samples to the testing phase makes the null-based thresholding highly variable: when $m$ is small, the test statistic exhibits large sampling fluctuations under $H_0$, so the resulting rejection threshold (or $p$-value) becomes noisy, which directly reduces power (Good, 2005). These limitations are intrinsic to split-based procedures and cannot be mitigated by increasing the size of the abundant reference sample alone. As a consequence, existing data-adaptive two-sample tests that rely on explicit train–test splitting may suffer from substantial power loss or unstable behavior in realistic scenarios where only a few query observations are available (Liu et al., 2020; Zhou et al., 2025b;a).

### 3.2. Can We Learn from Sufficient Reference Samples

A natural question arising from the limitations of train–test split-based methods is whether meaningful representations can be learned without access to query samples. This question has been extensively studied in the context of *one-class classification* (OCC), where models are trained exclusively on samples from a single distribution to characterize its support, geometry, and internal structure. Despite the absence of explicit negative examples, OCC methods have been empirically shown to learn informative representations that enable reliable detection of deviations at test time (Ruff et al., 2018; 2021; Chen et al., 2022).

This insight directly applies to the size-asymmetric two-sample testing settings considered in this work. When the reference distribution $\mathbb{P}$ is well-sampled, its intrinsic structure can be estimated reliably from reference data alone. Under the null hypothesis $\mathbb{P} = \mathbb{Q}$, query samples are expected to be compatible to these learned representations, whereas under alternatives $\mathbb{P} \neq \mathbb{Q}$, systematic deviations from the reference structure emerge. Crucially, learning representations solely from the reference sample avoids further splitting the already limited query set and eliminates a major source of instability in data-adaptive tests.

Moreover, no single reference-dependent representation can be expected to capture all possible forms of distributional discrepancy. Deviations between $\mathbb{P}$ and $\mathbb{Q}$ may manifest through changes in global moments, local density structure, or neighborhood geometry. This motivates the use of a family of complementary reference-dependent representations, each probing a distinct aspect of the reference distribution. Aggregating evidence across such a family enables robust detection of a broad class of alternatives while preserving the one-sided learning paradigm. These considerations motivate a reference-only learning framework that constructs and aggregates multiple reference-dependent representations to enable valid and powerful two-sample testing under size asymmetry while one sample is extremely limited.

## 4. Learning from Reference-Only Samples for Two-Sample Testing

Motivated by the limitations of split-based data-adaptive tests and the sufficiency of reference-only learning in Section 3, we formalize a one-sided framework for two-sample testing under severe size asymmetry. We consider the regime where the reference sample $X \sim \mathbb{P}$ is abundant while the query sample $Y \sim \mathbb{Q}$ is limited, a setting commonly encountered in practical applications (Li et al., 2021; Gao et al., 2021; Guille-Escuret et al., 2023; Li et al., 2025). We propose *Learning from Reference-Only Samples in Two-Sample Testing under Size Asymmetry (LOTTERY, or LOTT)*, a framework that constructs representations using only the abundant reference sample and performs testing by evaluating the compatibility of query samples with these representations. This framework avoids train–test splitting of the query data and enables valid level-$\alpha$ hypothesis testing through pooled permutation. Specifically, we partition the reference sample into three disjoint subsets $X = X^{\text{tr}} \cup X^{\text{cal}} \cup X^{\text{hold}}$, used respectively for learning one-sided representations, calibration, and pooled permutation testing.

### 4.1. Reference-Dependent Representation

Instead of directly learning a discrepancy between $\mathbb{P}$ and $\mathbb{Q}$ from a split of the two samples, we take a reference-only perspective. We first learn representations that characterize

the reference distribution $\mathbb{P}$ from samples, and then assess how well the query samples from $\mathbb{Q}$ are compatible with this reference. This view is closely related to OCC (Ruff et al., 2018), where a model is trained to capture the support of a single distribution and departures from it are treated as evidence against compatibility.

We formalize this idea through *reference-dependent representations* (RDR) defined as follows.

**Definition 4.1.** A reference-dependent representation is a function $f : \mathcal{X} \to \mathbb{R}$ learned exclusively from reference samples $X^{\mathrm{tr}} = \{\boldsymbol{x}_i^{\mathrm{tr}}\}_{i=1}^{n^{\mathrm{tr}}} \overset{i.i.d.}{\sim} \mathbb{P}$, such that larger values of $f(\boldsymbol{y})$ indicate that $\boldsymbol{y}$ is less compatible with $\mathbb{P}$.

An RDR maps each input to a scalar score, where larger values indicate lower compatibility with the reference distribution $\mathbb{P}$. Given a small query batch $Y = \{\boldsymbol{y}_j\}_{j=1}^m \overset{i.i.d.}{\to} \mathbb{Q}$, we test $\mathbb{P} = \mathbb{Q}$ by checking whether the query scores $\{f(\boldsymbol{y}_j)\}_{j=1}^m$ are statistically consistent with the reference score distribution induced by $\boldsymbol{x} \sim \mathbb{P}$.

### 4.2. Families of RDRs

We employ a set of complementary RDRs that capture different aspects of $\mathbb{P}$, including (i) RDRs for global structure and (ii) RDRs for local neighborhood structure. All RDRs below are learned exclusively from $X^{\mathrm{tr}}$. Furthermore, one feature is explicitly designed to guarantee consistency: for any alternative $\mathbb{P} \neq \mathbb{Q}$, the resulting two-sample test has power converging to one as the sample sizes grow, meaning that the probability of correctly rejecting the null hypothesis $H_0 : \mathbb{P} = \mathbb{Q}$ approaches one.

**Consistent ME-RDR for Local Similarity Patterns**. Inspired by Jitkrittum et al. (2016); Zhou et al. (2023), we introduce the *Mean Embedding (ME)-RDR* as a simple and consistent representation. Let $\kappa$ be a characteristic kernel on $\mathcal{X}$ and let $\{\boldsymbol{v}_\ell\}_{\ell=1}^L \subset X^{\mathrm{tr}}$ be a set of test locations selected from the reference sample. For each test location $\boldsymbol{v}_\ell$, define

$$f_\ell(\boldsymbol{x}) = \kappa(\boldsymbol{x}, \boldsymbol{v}_\ell), \qquad \ell = 1, \ldots, L.$$

Each $f_\ell$ measures the similarity between $\boldsymbol{x}$ and a fixed test location, thus capturing local deviations through how the query points align with $\mathbb{P}$ around $\boldsymbol{v}_\ell$.

*Remark 4.2.* For any test location $\boldsymbol{v} \in \mathcal{X}$, define $\mu_{\mathbb{P}}(\boldsymbol{v}) = \mathbb{E}_{\boldsymbol{x} \sim \mathbb{P}}[\kappa(\boldsymbol{x}, \boldsymbol{v})]$ and $\mu_{\mathbb{Q}}(\boldsymbol{v}) = \mathbb{E}_{\boldsymbol{y} \sim \mathbb{Q}}[\kappa(\boldsymbol{y}, \boldsymbol{v})]$. If $\kappa$ is characteristic and analytic and the test locations $\{\boldsymbol{v}_\ell\}_{\ell=1}^L$ are drawn i.i.d. from a distribution absolutely continuous with respect to the Lebesgue measure, then under any alternative $\mathbb{P} \neq \mathbb{Q}$ the ME-RDR family potentially yields a consistent test: as the sample sizes grow, the test power approaches one (Jitkrittum et al., 2016; Zhou et al., 2023).

**Mahalanobis-RDR for Global Structure**. Global RDRs measure deviations in coarse, distribution-level properties

of $\mathbb{P}$. A canonical example is the *Mahalanobis*-RDR

$$f_{\mathrm{Mah}}(\boldsymbol{x}) = (\boldsymbol{x} - \hat{\boldsymbol{\mu}})^\top \hat{\Sigma}^{-1} (\boldsymbol{x} - \hat{\boldsymbol{\mu}}),$$

where $\hat{\boldsymbol{\mu}}$ and $\hat{\Sigma}$ are the empirical mean and covariance computed from $X^{\mathrm{tr}}$. Such features are effective for detecting global mean and covariance shifts (Mahalanobis, 1936).

$k$**NN-RDR for Local Scale Geometry**. To capture fine-grained changes in local scale and support geometry, we use a nearest-neighbor RDR:

$$f_{\mathrm{kNN}}(\boldsymbol{x}) = \sum_{i=1}^k d_{(i)}(\boldsymbol{x})/k,$$

where $d(\cdot, \cdot)$ is a metric (Euclidean in our implementation) and $d_{(1)}(\boldsymbol{x}) \leq \cdots \leq d_{(k)}(\boldsymbol{x})$ are the ordered distances from $\boldsymbol{x}$ to its $k$ nearest neighbors in $X^{\mathrm{tr}}$. This score summarizes the local neighborhood radius around $\boldsymbol{x}$, making it sensitive to local sparsity/density changes and manifold- or cluster-type alternatives (Ramaswamy et al., 2000).

**LOF-RDR for Local Relative Density**. We also consider a *local outlier factor (LOF)-RDR* based on the LOF score (Breunig et al., 2000):

$$f_{\mathrm{LOF}}(\boldsymbol{x}) = \mathrm{median}_{\boldsymbol{y} \in N_k(\boldsymbol{x})} \frac{\mathrm{lrd}(\boldsymbol{y})}{\mathrm{lrd}(\boldsymbol{x})},$$

where $\mathrm{lrd}(\boldsymbol{x})$ is the local reachability density of $\boldsymbol{x}$ and $N_k(\boldsymbol{x})$ denotes its $k$ nearest neighbors in $X^{\mathrm{tr}}$. Unlike $k$NN distances, LOF compares the density of $\boldsymbol{x}$ to that of its neighbors, capturing local relative density irregularities, i.e., how outlying $\boldsymbol{x}$ is with respect to its surrounding region.

### 4.3. Aggregation from Multiple RDRs

Let $\mathcal{F}$ denote the set of all RDRs introduced in Section 4.2. Since different RDRs may operate on different scales, we standardize each RDR using the calibration set $X^{\mathrm{cal}} = \{\boldsymbol{x}_i\}_{i=1}^{n_{\mathrm{cal}}} \sim \mathbb{P}$. For each $f \in \mathcal{F}$, we define the calibration mean and variance

$$\bar{f} = \sum_{i=1}^{n_{\mathrm{cal}}} f(\boldsymbol{x}_i)/n_{\mathrm{cal}}, \qquad \hat{\sigma}_f^2 = \sum_{i=1}^{n_{\mathrm{cal}}} \big(f(\boldsymbol{x}_i) - \bar{f}\big)^2/(n_{\mathrm{cal}} - 1),$$

and the standardized score

$$a_f(\boldsymbol{x}) = (f(\boldsymbol{x}) - \bar{f})/\hat{\sigma}_f.$$

In practice, we add a small regularization term in the denominator when $\hat{\sigma}_f$ is close to zero to avoid numerical issues. This standardization places different RDRs on a comparable scale, so a large value of $a_f(\boldsymbol{x})$ indicates that $\boldsymbol{x}$ is atypical relative to the reference behavior captured by $f$.

Given a query batch $Y = \{\boldsymbol{y}_j\}_{j=1}^m \sim \mathbb{Q}$, we summarize the evidence provided by $f$ via squared mean standardized shift

$$T_f(Y) = \Big( \sum_{j=1}^m a_f(\boldsymbol{y}_j)/m \Big)^2.$$

Intuitively, under the null hypothesis $H_0 : \mathbb{P} = \mathbb{Q}$, the query points are drawn from same distribution as the calibration data, so the standardized query scores should fluctuate around zero and their average should be close to zero, making $T_f(Y)$ small. Under alternatives, if $f$ captures a direction along which $\mathbb{Q}$ systematically differs from $\mathbb{P}$, then the query scores tend to be shifted upward on average, and the squared mean shift $T_f(Y)$ becomes noticeably larger.

Finally, we aggregate evidence across all RDRs by summing their contributions:

$$T(Y) = \sum_{f \in \mathcal{F}} T_f(Y).$$

Large values of $T(Y)$ indicate that the query batch deviates from the reference distribution in one or more RDR directions, providing evidence against $H_0 : \mathbb{P} = \mathbb{Q}$.

### 4.4. Testing via Pooled Permutation

In the testing phase, we reserve a held-out reference set $X^{\text{hold}} = \{\boldsymbol{x}_i\}_{i=1}^{n_{\text{hold}}} \sim \mathbb{P}$ that is independent of $(X^{\text{tr}}, X^{\text{cal}})$. Given a query batch $Y = \{\boldsymbol{y}_j\}_{j=1}^m$, we form the pooled sample as follows

$$U = X^{\text{hold}} \cup Y,$$

which contains $n_{\text{hold}} + m$ points.

We compute the observed test statistic $T(Y)$ using the standardized scores. To test this statistic under the null, we use a pooled permutation (randomization) procedure: repeatedly draw a subset $S \subset U$ of size $m$ uniformly at random, treat $S$ as a pseudo-query batch, and compute the same statistic $T(S)$. Under $H_0 : \mathbb{P} = \mathbb{Q}$, all points in $U$ are i.i.d. from the same distribution, so the distinction between *query* and *held-out reference* is exchangeable (Lehmann & Romano, 2005; Gretton et al., 2012). Therefore, the distribution of $T(S)$ over uniformly random size-$m$ subsets provides a valid calibration for $T(Y)$ as shown in Theorem 6.1.

In practice, we approximate the permutation threshold using $B$ random subsets. Let $S_1, \ldots, S_B$ be i.i.d. uniformly sampled size-$m$ subsets of $U$, and compute $\{T(S_b)\}_{b=1}^B$. Given a significance level $\alpha \in (0, 1)$, we define the testing threshold as the empirical $(1 - \alpha)$-quantile of the multiset $\{T(S_1), \ldots, T(S_B), T(Y)\}$:

$$\hat{\tau}_\alpha = \inf \left\{ t \in \mathbb{R} : \frac{1}{B+1} \sum_{b=1}^{B+1} \mathbf{1}\{T(S_b) \leq t\} \geq 1 - \alpha \right\},$$

where we set $S_{B+1} = Y$. The resulting level-$\alpha$ test is

$$\phi(X, Y) = \mathbf{1}\{T(Y) > \hat{\tau}_\alpha\}.$$

## 5. Selection Based on Uncertainty Weighting

The performance of a two-sample test is governed by a relative comparison between two quantities: **the discrepancy signal** and **the testing threshold**. The discrepancy signal is the observed value of the test statistic on the query batch, which becomes larger when $\mathbb{P}$ and $\mathbb{Q}$ differ more in the direction probed by the statistic. The testing threshold is the $(1 - \alpha)$-quantile of the statistic under $H_0 : \mathbb{P} = \mathbb{Q}$, obtained from its null (permutation) distribution; it is larger when the statistic fluctuates more under $H_0$ (i.e., has higher null variability). Intuitively, larger discrepancy increases the statistic, whereas higher null variability inflates the threshold, making rejection harder. Therefore, test power improves when the statistic exhibits a strong discrepancy signal while remaining stable under the null.

Our framework deliberately avoids train-test splitting and does not learn representations to explicitly amplify the (unknown) discrepancy between $\mathbb{P}$ and $\mathbb{Q}$. As a consequence, we cannot directly tune the test statistic to increase **the discrepancy signal** under alternatives. What we *can* control is **the testing threshold**: since the RDRs and their standardization are learned from the reference data, the null behavior of the aggregated statistic can be assessed using only reference samples via pooled permutation, and the threshold is largely driven by how much the statistic fluctuates when we form pseudo-query batches from $\mathbb{P}$. This motivates an uncertainty-aware aggregation strategy: we use resampling on the reference sample to quantify the stability of each RDR, and then weight or select RDRs to downweight unstable ones. This directly reduces the null variability of the aggregated statistic, leading to tighter permutation thresholds and improved power under sample-size asymmetry.

To quantify the uncertainty of a RDR $f \in \mathcal{F}$, we measure how much its batch mean fluctuates when we resample from the calibration set $X^{\text{cal}} = \{\boldsymbol{x}_i\}_{i=1}^{n_{\text{cal}}} \sim \mathbb{P}$. Concretely, draw random subsets $\mathcal{S}_1, \ldots, \mathcal{S}_R \subset X^{\text{cal}}$ of a size $m$, and define

$$\bar{a}_f^r = \sum_{\boldsymbol{x} \in \mathcal{S}_r} a_f(\boldsymbol{x})/|\mathcal{S}_r|, \qquad r = 1, \ldots, R,$$

where $a_f(\cdot)$ is the standardized score defined in Section 4.2. We estimate the uncertainty of $f$ by the empirical variance

$$\hat{\sigma}_{a_f}^2 = \sum_{r=1}^R \left( \bar{a}_f^r - \frac{1}{R} \sum_{r'=1}^R \bar{a}_f^r \right)^2 / (R-1).$$

Inspired by Kendall et al. (2018), this quantity serves as a reference-based stability measure: a larger $\hat{\sigma}_{a_f}^2$ indicates that the RDR mean is sensitive to small perturbations of the reference sample, and such unstable RDRs tend to inflate the variability of the aggregated test statistic.

**Uncertainty Alone Is Insufficient.** Uncertainty measures stability, but it does not measure usefulness for detection. For example, a degenerate RDR that outputs an (almost) constant score has nearly zero variance and would be favored by variance-based weighting, yet it carries no information

about distributional differences. Therefore, weighting RDRs only by inverse uncertainty may overemphasize stable but uninformative representations and reduce power.

**Sensitivity-Aware Uncertainty Weighting.** To address this, we couple stability with a simple notion of sensitivity. The key idea is that a useful RDR should be stable when evaluated on resampled reference data, but should also respond to small, controlled departures from the reference distribution. Specifically, let $\widetilde{X}^{\text{cal}}$ be a mildly perturbed version of the calibration set, constructed by applying a small perturbation to the reference samples. For each $f \in \mathcal{F}$, we define a sensitivity score

$$\widehat{\delta}_{a_f} = \Big| \sum_{\boldsymbol{y} \in \widetilde{X}^{\text{cal}}} a_f(\boldsymbol{y}) - \sum_{\boldsymbol{x} \in X^{\text{cal}}} a_f(\boldsymbol{x}) \Big| / n_{\text{cal}},$$

which quantifies how much the standardized scores change under a small perturbation. RDRs with $\widehat{\delta}_{a_f}$ close to zero are largely insensitive and are unlikely to contribute power.

**Combined Weighting Scheme.** We combine stability and sensitivity by setting

$$w_f = \widehat{\delta}_{a_f} / \widehat{\sigma}_{a_f}^2 \, ,$$

and throughout we assume $\widehat{\delta}_{a_f} > 0$ and $\widehat{\sigma}_{a_f}^2 > 0$ so that the weight is well-defined.

This weight favors RDRs that are both stable (small $\widehat{\sigma}_{a_f}^2$) and responsive (large $\widehat{\delta}_{a_f}$). Finally, we aggregate evidence across all RDRs by weighting their contributions:

$$T_{\text{weight}}(Y) = \sum_{f \in \mathcal{F}} w_f \, T_f(Y).$$

Downweighting unstable RDRs reduces the null variability of the aggregated statistic and leads to tighter permutation thresholds, while incorporating sensitivity prevents trivial but stable RDRs from dominating. Together, these choices stabilize permutation calibration and can improve power in finite-sample, imbalanced regimes, while using only reference data and retaining validity under $H_0$.

## 6. Theoretical Analysis

We first show that the proposed pooled-permutation procedure is valid under the null hypothesis, providing finite-sample type I error control.

**Theorem 6.1.** *Let $X^{\text{hold}} \sim \mathbb{P}^{n_{\text{hold}}}$ and $Y \sim \mathbb{Q}^m$, independent of $(X^{\text{tr}}, X^{\text{cal}})$. Then, under null $H_0 : \mathbb{P} = \mathbb{Q}$, the permutation test controls the type I error at level $\alpha$.*

We next establish consistency under alternative hypothesis.

**Theorem 6.2.** *Fix $\alpha \in (0, 1)$ and $B \geq 1$ such that $1/(B + 1) < \alpha$. Let $X^{\text{hold}} \sim \mathbb{P}^{n_{\text{hold}}}$ and $Y \sim \mathbb{Q}^m$, independent of $(X^{\text{tr}}, X^{\text{cal}})$. Let $n_{\text{hold}} = n_{\text{hold},m}$ with $m \to \infty$*

*and assume $n_{\text{hold},m}/m \to \rho \in (0, \infty)$. Let $\{w_f\}_{f \in \mathcal{F}}$ be positive weights constructed from $(X^{\text{tr}}, X^{\text{cal}})$ (including $w_f = 1$, for which $T_{\text{weight}}(Y) = T(Y)$). Assume $\mathbb{E}[a_f(X)^2] < \infty$ and $\mathbb{E}[a_f(Y)^2] < \infty$ for all $f \in \mathcal{F}$. If $n_{\text{hold},m} \neq o(m)$ and there exists $f \in \mathcal{F}$ that is consistent, then the pooled permutation test with aggregated statistic is consistent, i.e., $\mathbb{P}\big(T_{\text{weight}}(Y) > \hat{\tau}_\alpha\big) \to 1$ as $m \to \infty$.*

Together, Theorems 6.1-6.2 show that our test is both valid under $H_0$ and consistent under $H_1$.

## 7. Experiments

In this section, we empirically evaluate the proposed method under a range of controlled and real-world settings. Our experiments are designed to answer three core questions: (i) whether the proposed approach is effective in the *extreme asymmetric setting* where the query sample size is severely limited; (ii) whether it exhibits *consistent power improvement* as the number of query samples increases; and (iii) whether the proposed uncertainty-based selection mechanism contributes meaningfully to stability and power.

### 7.1. Datasets

We consider both synthetic benchmarks commonly used in two-sample testing and realistic large-scale datasets that reflect practical detection scenarios. The details of datasets generation and shared embedding extraction can be found in the Appendix B.1

**Synthetic Data.** We use the standard BLOB dataset (Gretton et al., 2012; Liu et al., 2020; Kübler et al., 2022; Schrab et al., 2023), which consists of mixtures of Gaussian components arranged on a grid. This dataset is widely adopted in the two-sample testing literature to evaluate sensitivity to local density and structural changes. We follow the standard experimental protocol used in prior work, varying the separation and covariance structure between distributions while controlling sample sizes.

**Physical Higgs Boson.** The Higgs dataset is a high-dimensional tabular dataset originating from particle physics (Baldi et al., 2014), where the task is to distinguish simulated signal events from background events. Following prior two-sample testing studies, we treat the background distribution as the reference and test against shifted or reweighted signal distributions. This dataset reflects realistic challenges such as heterogeneous features and moderate dimensionality.

**Adversarial Detection on CIFAR-10.** To evaluate performance in high-dimensional image settings, we consider adversarial detection on CIFAR-10 (Krizhevsky & Hinton, 2009). Clean images are treated as reference samples, while adversarially perturbed images form the query distribution. We consider multiple threat models and architectures,

*Table 1.* Test power (mean$_\pm$std) on BLOB and CIFAR10-RES18 with fixed $M$ and varying sample size $N$. Standard deviation is reported without a leading zero. Best results are bolded. $\Delta$ reports the absolute difference between LOTTERY and the second-best method in each column (+: outperform, −: underperform). The upper section of baselines are kernel-based methods, and the lower section of baselines are learning-based methods.

| Method | BLOB ($M = 50$) | | | | | | | CIFAR10-RES18 ($M = 4$) | | | | | | |
|---|---|---|---|---|---|---|---|---|---|---|---|---|---|---|
| | $N=100$ | $N=150$ | $N=200$ | $N=300$ | $N=400$ | $N=500$ | $N=1000$ | $N=40$ | $N=50$ | $N=60$ | $N=70$ | $N=80$ | $N=90$ | $N=100$ |
| MMD-FUSE | **0.122**$_{\pm.011}$ | **0.146**$_{\pm.011}$ | **0.162**$_{\pm.011}$ | 0.138$_{\pm.011}$ | 0.163$_{\pm.009}$ | 0.153$_{\pm.010}$ | 0.145$_{\pm.008}$ | 0.280$_{\pm.015}$ | 0.282$_{\pm.011}$ | 0.290$_{\pm.014}$ | 0.303$_{\pm.013}$ | 0.302$_{\pm.015}$ | 0.288$_{\pm.015}$ | 0.303$_{\pm.017}$ |
| MMDAgg | 0.060$_{\pm.007}$ | 0.053$_{\pm.003}$ | 0.077$_{\pm.010}$ | 0.046$_{\pm.005}$ | 0.068$_{\pm.009}$ | 0.056$_{\pm.008}$ | 0.046$_{\pm.006}$ | 0.283$_{\pm.013}$ | 0.291$_{\pm.012}$ | 0.287$_{\pm.011}$ | 0.285$_{\pm.012}$ | 0.307$_{\pm.009}$ | 0.292$_{\pm.012}$ | 0.317$_{\pm.019}$ |
| MMD-M | 0.059$_{\pm.007}$ | 0.064$_{\pm.006}$ | 0.064$_{\pm.006}$ | 0.059$_{\pm.006}$ | 0.054$_{\pm.008}$ | 0.056$_{\pm.007}$ | 0.056$_{\pm.006}$ | 0.483$_{\pm.017}$ | 0.501$_{\pm.008}$ | 0.504$_{\pm.013}$ | 0.508$_{\pm.014}$ | 0.521$_{\pm.014}$ | 0.521$_{\pm.011}$ | 0.531$_{\pm.023}$ |
| MMD-Deep | 0.076$_{\pm.007}$ | 0.078$_{\pm.010}$ | 0.096$_{\pm.012}$ | 0.079$_{\pm.009}$ | 0.087$_{\pm.012}$ | 0.066$_{\pm.007}$ | 0.073$_{\pm.010}$ | 0.200$_{\pm.015}$ | 0.224$_{\pm.023}$ | 0.216$_{\pm.015}$ | 0.209$_{\pm.024}$ | 0.215$_{\pm.020}$ | 0.241$_{\pm.029}$ | 0.263$_{\pm.020}$ |
| RL-TST | 0.070$_{\pm.008}$ | 0.113$_{\pm.014}$ | 0.105$_{\pm.012}$ | 0.109$_{\pm.017}$ | 0.178$_{\pm.021}$ | 0.193$_{\pm.025}$ | 0.180$_{\pm.023}$ | 0.070$_{\pm.018}$ | 0.060$_{\pm.017}$ | 0.078$_{\pm.011}$ | 0.060$_{\pm.018}$ | 0.091$_{\pm.016}$ | 0.150$_{\pm.025}$ | 0.189$_{\pm.014}$ |
| C2ST-L | 0.072$_{\pm.015}$ | 0.073$_{\pm.014}$ | 0.074$_{\pm.008}$ | 0.087$_{\pm.021}$ | 0.070$_{\pm.008}$ | 0.128$_{\pm.030}$ | 0.048$_{\pm.010}$ | 0.076$_{\pm.014}$ | 0.058$_{\pm.008}$ | 0.086$_{\pm.017}$ | 0.081$_{\pm.018}$ | 0.187$_{\pm.023}$ | 0.201$_{\pm.026}$ | 0.132$_{\pm.012}$ |
| IT | 0.082$_{\pm.006}$ | 0.139$_{\pm.013}$ | 0.157$_{\pm.007}$ | **0.237**$_{\pm.013}$ | **0.414**$_{\pm.018}$ | **0.579**$_{\pm.014}$ | **0.725**$_{\pm.015}$ | **0.789**$_{\pm.013}$ | **0.822**$_{\pm.010}$ | **0.874**$_{\pm.006}$ | **0.893**$_{\pm.010}$ | **0.917**$_{\pm.006}$ | **0.891**$_{\pm.007}$ | **0.932**$_{\pm.005}$ |
| $\Delta$ (vs. 2nd best) | −0.040 | −0.007 | −0.005 | +0.099 | +0.236 | +0.386 | +0.545 | +0.306 | +0.321 | +0.370 | +0.385 | +0.396 | +0.370 | +0.401 |

including ResNet-18 (He et al., 2016) and WideResNet-28 (Zagoruyko & Komodakis, 2016), and adopt commonly used adversarial attack methods, in particular projected gradient descent (PGD) (Madry et al., 2018) with adversarial budget $\epsilon = 4/255$. Embedding representations are extracted from the penultimate (global average pooled) layer of the threat model, and all two-sample testing methods operate on the same embedding space for fair comparison.

## 7.2. Baselines

We compare the proposed method against representative state-of-the-art non-parametric two-sample testing methods, including both kernel-based and learning-based approaches:

- **C2ST-L** (Lopez-Paz & Oquab, 2017; Kim et al., 2021): classifier two-sample test (C2ST) based on training a binary classifier and using its logits as a proxy for MMD.

- **MMD-M** (Garreau et al., 2017): MMD with a single Gaussian kernel using the median heuristic for bandwidth.

- **MMD-FUSE** (Biggs et al., 2023) and **MMDAgg** (Schrab et al., 2023): multi-kernel aggregation methods designed to capture information across various kernel choices while preserving type I error control without data splitting.

- **MMD-Deep** (Liu et al., 2020) and **RL-TST** (Tian et al., 2025): deep-kernel MMD with representation learning on a train set and testing on a held-out set. The representation can either be supervised or self-supervised learned.

All baselines are carefully tuned following the recommendations in their original papers, and permutation testing is used whenever applicable to ensure valid type I error control.

## 7.3. Results & Analysis

We evaluate test performance primarily in terms of empirical power at a fixed significance level $\alpha = 0.05$, averaged over 10 independent trials, in each trial, the two samples will be drawn 100 times and each time will report a null hypothesis rejection or not. Moreover, we conduct the type I error check with $\alpha$ ranging from 0.05 to even 0.01, showing the

control and validity of type I error. In the main content, we display the results of different experimental settings on dataset BLOB and CIFAR10-ResNet18, all the rest of the results can be found in Appendix B.1.

**Varying the reference size $N$ (few-shot power).** We investigate how effectively LOTTERY exploits an increasing number of reference samples when the query size $M$ is fixed and small. Results are summarized in Table 1. In the low-$N$ regime, where the reference set is not abundant enough compared to the query size $M$, LOTTERY can slightly underperform the strongest kernel baselines, reflecting the limited information available for learning a RDR. However, as $N$ increases, LOTTERY exhibits a markedly faster power growth than all baselines. On BLOB ($M{=}50$), LOTTERY overtakes every competing method from $N \geq 300$ onward and continues to widen the margin as more reference data become available. In contrast, kernel-based tests and split-based learning baselines show much slower improvements with increasing $N$, indicating limited scalability with respect to the reference size. On CIFAR10-RES18 ($M{=}4$), LOTTERY achieves consistently high power across all $N$ and substantially outperforms both kernel and learning-based baselines, even when the reference set is small. These results highlight that while LOTTERY may not dominate in extremely low-reference regimes, it benefits disproportionately from additional reference samples, making it particularly effective in few-query, abundant-reference settings.

**Varying the query size $M$ (empirical consistency).** We next examine how test power evolves as the number of query samples increases while keeping the reference set fixed ($N{=}4000$ for BLOB and $N{=}100$ for CIFAR10-RES18); see Table 2. This setting represents an extreme asymmetry regime, where only a handful of query samples are available against a large reference pool. Across both datasets, LOTTERY consistently achieves the highest power for all $M$, with particularly pronounced advantages in the most few-shot regimes. For instance, on CIFAR10-RES18 with only $M{=}2$ queries, LOTTERY already attains strong detection performance, whereas kernel-based tests and split-based learning baselines remain close to chance or are not ap-

*Table 2.* Test power (mean$_\pm$std) on BLOB and CIFAR10-RES18 with fixed $N$ and varying sample size $M$. Standard deviation is reported without a leading zero. Best results are bolded. $\Delta$ reports the absolute difference between LOTTERY and the second-best method in each column (+: outperform, −: underperform). The upper section of baselines are kernel-based methods, and the lower section of baselines are learning-based methods. MMD-Deep cannot work when $M=2$.

| Method | BLOB ($N=4000$) | | | | | | | CIFAR10-RES18 ($N=100$) | | | | | | |
|---|---|---|---|---|---|---|---|---|---|---|---|---|---|---|
| | $M=20$ | $M=50$ | $M=80$ | $M=100$ | $M=150$ | $M=200$ | $M=300$ | $M=2$ | $M=4$ | $M=6$ | $M=8$ | $M=10$ | $M=12$ | $M=14$ |
| MMD-FUSE | $0.056_{\pm.008}$ | $0.120_{\pm.011}$ | $0.269_{\pm.010}$ | $0.414_{\pm.009}$ | $0.776_{\pm.012}$ | $0.919_{\pm.009}$ | $1.000_{\pm.000}$ | $0.086_{\pm.006}$ | $0.304_{\pm.009}$ | $0.488_{\pm.014}$ | $0.723_{\pm.017}$ | $0.853_{\pm.012}$ | $0.923_{\pm.007}$ | $0.975_{\pm.004}$ |
| MMDAgg | $0.025_{\pm.006}$ | $0.065_{\pm.008}$ | $0.121_{\pm.013}$ | $0.177_{\pm.013}$ | $0.381_{\pm.013}$ | $0.483_{\pm.016}$ | $0.598_{\pm.015}$ | $0.097_{\pm.007}$ | $0.305_{\pm.011}$ | $0.405_{\pm.014}$ | $0.549_{\pm.018}$ | $0.576_{\pm.016}$ | $0.612_{\pm.013}$ | $0.668_{\pm.010}$ |
| MMD-M | $0.049_{\pm.009}$ | $0.058_{\pm.006}$ | $0.055_{\pm.006}$ | $0.064_{\pm.005}$ | $0.058_{\pm.006}$ | $0.054_{\pm.004}$ | $0.071_{\pm.007}$ | $0.146_{\pm.012}$ | $0.519_{\pm.013}$ | $0.713_{\pm.015}$ | $0.868_{\pm.007}$ | $0.945_{\pm.008}$ | $0.974_{\pm.004}$ | $0.991_{\pm.003}$ |
| MMD-Deep | $0.081_{\pm.007}$ | $0.087_{\pm.008}$ | $0.087_{\pm.011}$ | $0.134_{\pm.023}$ | $0.316_{\pm.040}$ | $0.350_{\pm.052}$ | $0.622_{\pm.086}$ | $-_{\pm.-}$ | $0.206_{\pm.017}$ | $0.251_{\pm.014}$ | $0.527_{\pm.030}$ | $0.561_{\pm.031}$ | $0.608_{\pm.035}$ | $0.662_{\pm.036}$ |
| RL-TST | $0.118_{\pm.013}$ | $0.226_{\pm.025}$ | $0.326_{\pm.036}$ | $0.067_{\pm.015}$ | $0.251_{\pm.077}$ | $0.453_{\pm.086}$ | $0.742_{\pm.079}$ | $0.105_{\pm.014}$ | $0.102_{\pm.014}$ | $0.171_{\pm.032}$ | $0.406_{\pm.035}$ | $0.430_{\pm.044}$ | $0.479_{\pm.038}$ | $0.568_{\pm.033}$ |
| C2ST-L | $0.064_{\pm.007}$ | $0.085_{\pm.019}$ | $0.046_{\pm.006}$ | $0.055_{\pm.007}$ | $0.042_{\pm.009}$ | $0.065_{\pm.006}$ | $0.055_{\pm.007}$ | $0.061_{\pm.010}$ | $0.068_{\pm.013}$ | $0.091_{\pm.017}$ | $0.328_{\pm.022}$ | $0.286_{\pm.053}$ | $0.423_{\pm.064}$ | $0.458_{\pm.045}$ |
| **LOTTERY** | $\mathbf{0.524}_{\pm.013}$ | $\mathbf{0.824}_{\pm.008}$ | $\mathbf{0.900}_{\pm.008}$ | $\mathbf{0.939}_{\pm.010}$ | $\mathbf{0.993}_{\pm.002}$ | $\mathbf{0.999}_{\pm.001}$ | $\mathbf{1.000}_{\pm.000}$ | $\mathbf{0.810}_{\pm.011}$ | $\mathbf{0.966}_{\pm.004}$ | $\mathbf{0.991}_{\pm.002}$ | $\mathbf{0.998}_{\pm.002}$ | $\mathbf{1.000}_{\pm.000}$ | $\mathbf{1.000}_{\pm.000}$ | $\mathbf{1.000}_{\pm.000}$ |
| $\Delta$ (vs. 2nd best) | +0.406 | +0.598 | +0.574 | +0.525 | +0.217 | +0.080 | +0.000 | +0.664 | +0.447 | +0.278 | +0.130 | +0.055 | +0.026 | +0.009 |

*Table 3.* Ablation of LOTTERY selection. Entries report test power as mean$_\pm$std, and $\Delta$ denotes the absolute gain from selection. The seven configurations $m_1$–$m_7$ follow the same choices of $M$ as in Table 2. $N=4000$ for BLOB and $N=100$ for CIFAR10.

| Dataset | Variant | $m_1$ | $m_2$ | $m_3$ | $m_4$ | $m_5$ | $m_6$ | $m_7$ |
|---|---|---|---|---|---|---|---|---|
| **BLOB** | LOTTERY | $0.524_{\pm.013}$ | $0.824_{\pm.008}$ | $0.900_{\pm.008}$ | $0.939_{\pm.010}$ | $0.993_{\pm.002}$ | $0.999_{\pm.001}$ | $1.000_{\pm.000}$ |
| | w/o sel. | $0.411_{\pm.011}$ | $0.661_{\pm.010}$ | $0.780_{\pm.012}$ | $0.836_{\pm.008}$ | $0.935_{\pm.003}$ | $0.979_{\pm.005}$ | $0.995_{\pm.003}$ |
| | $\Delta$ | +0.113 | +0.163 | +0.120 | +0.103 | +0.058 | +0.020 | +0.005 |
| **CIFAR10** | LOTTERY | $0.810_{\pm.011}$ | $0.966_{\pm.004}$ | $0.991_{\pm.002}$ | $0.998_{\pm.002}$ | $1.000_{\pm.000}$ | $1.000_{\pm.000}$ | $1.000_{\pm.000}$ |
| | w/o sel. | $0.768_{\pm.012}$ | $0.916_{\pm.006}$ | $0.968_{\pm.006}$ | $0.987_{\pm.003}$ | $0.994_{\pm.002}$ | $0.997_{\pm.001}$ | $0.999_{\pm.001}$ |
| | $\Delta$ | +0.042 | +0.050 | +0.023 | +0.011 | +0.006 | +0.003 | +0.001 |

plicable. As $M$ increases, the performance gap gradually narrows as competing methods benefit from additional query information; nevertheless, LOTTERY maintains a clear lead in the low-to-moderate $M$ range and rapidly approaches near-perfect power. This behavior indicates that LOTTERY is especially effective under severe query scarcity, while remaining empirically consistent and converging to 1.

**Ablation Study.** Moreover, we conduct an ablation study to assess the contribution of the proposed *uncertainty-based selection and weighting* mechanism. Table 3 isolates the contribution of our uncertainty-guided selection. Selection yields consistent gains across all $m$ values, with the largest improvements when $m$ is smallest (e.g., +0.113 on BLOB at $M=20$ and +0.050 on CIFAR10-RES18 at $M=4$). This supports our motivation in Section 5: downweighting unstable RDRs tightens the threshold and improves power.

**Complementarity of RDR families.** We further evaluate why aggregating multiple RDR families is useful. In this experiment, we construct four synthetic alternatives with the same reference distribution $P = \mathcal{N}(0, I_d)$, where $d = 10$. The alternatives are designed to emphasize different types of distributional changes, including global location shifts, global scale changes, heterogeneous coordinate-wise variance changes, and localized point contamination. Table 4 reports a compact comparison at $m = 50$, with the full data generation details and results over $m \in \{20, 50, 100, 200\}$ provided in Appendix B.2. No single RDR is uniformly best: Mahalanobis-RDR performs strongly on global mean and variance changes, whereas LOF-RDR becomes competitive on heterogeneous variance changes. This confirms that the RDR families capture non-redundant aspects of the

*Table 4.* Complementarity of individual RDR families. We report test power at $\alpha = 0.05$ with $N = 4000$ reference samples and $M = 50$ query samples. Bold numbers indicate the best individual RDR in each row. Full results over all query sizes and how the alternative synthetic data generated are provided in Appendix B.2.

| Alternative | LOF | kNN | Mahalanobis | LOTTERY |
|---|---|---|---|---|
| mean_shift | .278 | .288 | **.320** | **.442** |
| variance_scale | .301 | .299 | **.338** | **.444** |
| skew_variance | **.447** | .403 | .444 | **.536** |
| point_contam | .203 | .219 | **.270** | **.317** |

reference distribution. By aggregating these complementary signals, LOTTERY outperforms every individual RDR across all alternatives, supporting the motivation for using multiple reference-dependent representations.

**Type I Error Check.** Figure 1 reports the empirical type I error of LOTTERY across datasets and $M$ settings, where $m_1$–$m_7$ follow the same choices of $M$ as in Table 2. Across all datasets, the type I error stays close to the significance level $\alpha = 0.05$. This demonstrates stable false-positive control under different configurations.

### 7.4. Discussion

The empirical results highlight three main insights. First, LOTTERY is most effective in the regime it is designed for: severe sample-size asymmetry with few query samples and abundant reference data. In this setting, split-based learning methods suffer from unstable representation learning and reduced testing power, whereas LOTTERY uses the query batch only for testing and exploits the reference sample for representation construction. Second, the gains

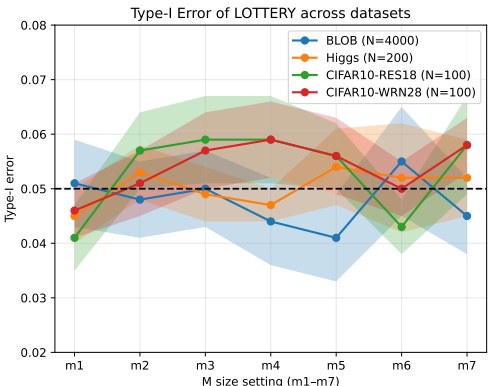

*Figure 1.* Type I error of the LOTTERY test across different datasets and $M$. The horizontal axis corresponds to seven configurations of the test statistic (denoted as $m_1$–$m_7$), while the vertical axis reports the type I error. Solid curves show the mean type I error over repeated trials, and shaded regions indicate standard deviation. Results are shown for BLOB ($N{=}4000$), Higgs ($N{=}200$), CIFAR10-RES18 ($N{=}100$), and CIFAR10-WRN28 ($N{=}100$). The black dashed line marks the significance level $\alpha{=}0.05$.

of LOTTERY come not only from using one-sided scores, but from converting a family of reference-dependent scores into a valid two-sample test with finite-sample type I error control. This distinguishes our setting from standard one-class classification or anomaly detection, where scores are often useful for ranking unusual samples but do not by themselves provide level-$\alpha$ two-sample testing guarantees. Third, different RDR families capture different aspects of the reference distribution. The complementarity results show that no single RDR is uniformly best across alternatives, while the uncertainty-guided aggregation improves power by combining stable and informative signals. These observations suggest that reference-only learning provides a practical and principled direction for future two-sample tests under extreme class imbalance.

## 8. Conclusion

We studied data-adaptive two-sample testing under severe sample-size asymmetry, where abundant reference data are available but only a few query samples are observed. We proposed LOTTERY, a reference-only testing framework that learns multiple reference-dependent representations, aggregates their evidence via uncertainty-guided selection, and calibrates the test using pooled permutation, thereby avoiding query data splitting while preserving nonparametric validity. We established finite-sample type I error control under $H_0$ and consistency under $H_1$ when the representation family contains at least one consistent component. Empirically, LOTTERY consistently outperforms kernel-based and learning-based baselines, with the largest gains in the most few-shot regimes, and ablations confirm the critical role of uncertainty-guided selection. These results demonstrate that reference-only learning is a principled and effective approach for two-sample testing under extreme imbalance.

## Acknowledgements

This research was supported by The University of Melbourne's Research Computing Services and the Petascale Campus Initiative. XYT is supported by Maincode research program, the Melbourne Research Scholarship and the ARC with grant number DE240101089. ZJZ is supported by the Melbourne Research Scholarship and the ARC with grant number DE240101089. LHP is supported by the ARC with grant number LP240100101. FL is supported by the ARC with grant number DE240101089, LP240100101, DP230101540 and the NSF&CSIRO Responsible AI program with grant number 2303037.

## Impact Statement

This work studies data-adaptive two-sample testing under severe sample-size asymmetry, a setting that arises naturally in applications such as anomaly detection, adversarial example detection, and monitoring distributional shifts in deployed machine-learning systems. By enabling valid and powerful testing when only a handful of query samples are available, the proposed method may help practitioners detect harmful or unexpected changes more reliably, potentially improving the safety and robustness of data-driven systems.

The proposed methodology is statistical in nature and does not introduce new mechanisms for data collection, surveillance, or decision-making about individuals. As with other hypothesis-testing tools, misuse may arise if results are interpreted without consideration of assumptions, uncertainty, or domain context. We therefore emphasize that LOTTERY is intended as a diagnostic component within a broader analytical pipeline rather than a standalone decision rule.

Overall, we believe this work has the potential for positive impact by strengthening principled distribution-shift detection in high-stakes settings, while posing minimal risk when applied responsibly.

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

# A. Proofs of Theoretical Analysis

## A.1. Detailed Proofs of Theorem 6.1

We begin with a useful definition as follows.

**Definition A.1.** (Hemerik & Goeman, 2018) Let $Z$ be the sample taking values in the instance space $\mathcal{X}$. Let $\mathcal{G}$ be a finite set of transformations $g : \mathcal{X} \to \mathcal{X}$, such that $\mathcal{G}$ is a group with respect to the operation of composition of transformations. Let $\mathcal{H}_0$ be any null hypothesis which implies that the joint distribution of the test statistics $T(gZ)$, $g \in \mathcal{G}$, is invariant under all transformations in $\mathcal{G}$ of $Z$. Denote by $R$ the cardinality of the set $\mathcal{G}$ of $\mathcal{G}$ and write $\mathcal{G} = \{g_1, ..., g_R\}$. We have, under $\mathcal{H}_0$,

$$(T(g_1 Z), ..., T(g_R Z)) \overset{d}{=} (T(g \cdot g_1 Z), ..., T(g \cdot g_R Z)) \quad \text{for all } g \in \mathcal{G} ,$$

where $\overset{d}{=}$ denotes equality in distribution.

We present the proofs of Theorem 6.1 as follows.

*Proof.* Let $X^{\text{hold}} = \{x_i\}_{i=1}^{n_{\text{hold}}}$ and $Y = \{y_j\}_{j=1}^m$, and form the pooled sample

$$U = X^{\text{hold}} \cup Y$$

with $N = n_{\text{hold}} + m$ points. Fix an arbitrary ordering of $U$ as

$$Z = (Z_1, \ldots, Z_N) = (x_1, \ldots, x_{n_{\text{hold}}}, y_1, \ldots, y_m).$$

We emphasize that the proof applies to both the unweighted statistic $T(Y) = \sum_{f \in \mathcal{F}} T_f(Y)$ and the weighted statistic $T_{\text{weight}}(Y) = \sum_{f \in \mathcal{F}} w_f T_f(Y)$. Indeed, once $(X^{\text{tr}}, X^{\text{cal}})$ are fixed, the standardization parameters and the weights $\{w_f\}$ are fixed, and the rule that maps any candidate batch $S$ to a scalar value $T(S)$ is deterministic. Since $X^{\text{hold}}$ and $Y$ are independent of $(X^{\text{tr}}, X^{\text{cal}})$, this deterministic rule does not depend on how the pooled points are labeled as held-out reference versus query.

Under $H_0 : \mathbb{P} = \mathbb{Q}$, we have $Z_1, \ldots, Z_N \overset{i.i.d.}{\sim} \mathbb{P}$, hence $Z$ is exchangeable with respect to the permutation group $\mathbf{\Pi}_N$. In the pooled permutation procedure, each replicate is obtained by drawing a size-$m$ subset $S \subset U$ uniformly at random and computing $T(S)$. Equivalently, draw a uniform random permutation $\mathbf{\Pi} \sim \text{Unif}(\mathbf{\Pi}_N)$ and let $S(\mathbf{\Pi})$ be the last $m$ elements of $\mathbf{\Pi} Z$; then $S(\mathbf{\Pi})$ is a uniformly random size-$m$ subset of $U$, and the replicate statistic can be written as $T(S(\mathbf{\Pi}))$.

Let $\mathbf{\Pi}_{(1)}, \ldots, \mathbf{\Pi}_{(B)} \overset{i.i.d.}{\sim} \text{Unif}(\mathbf{\Pi}_N)$ be independent of $Z$, define $S_b = S(\mathbf{\Pi}_{(b)})$, and write

$$T_{B+1} = T(Y), \qquad T_b = T(S_b), \quad b \in [B].$$

Let $T_{(1)} \leq \cdots \leq T_{(B+1)}$ be the order statistics of the multiset $\{T_1, T_2, \ldots, T_{B+1}\}$. Define the empirical $(1 - \alpha)$-quantile threshold by

$$\hat{\tau}_\alpha = T_{(k)}, \qquad k = \lceil (1 - \alpha)(B + 1) \rceil ,$$

and reject $H_0$ if $T_0 > \hat{\tau}_\alpha$.

To show type I error control, introduce an additional independent $\mathbf{\Pi} \sim \text{Unif}(\mathbf{\Pi}_N)$. By exchangeability of $Z$ under $H_0$ and the group property of $\mathbf{\Pi}_N$, we have

$$\big(T(S(\text{Id})), T(S(\mathbf{\Pi}_{(1)})), \ldots, T(S(\mathbf{\Pi}_{(B)}))\big) \overset{d}{=} \big(T(S(\mathbf{\Pi})), T(S(\mathbf{\Pi}\mathbf{\Pi}_{(1)})), \ldots, T(S(\mathbf{\Pi}\mathbf{\Pi}_{(B)}))\big),$$

where Id is the identity permutation. Conditionally on $(Z, \mathbf{\Pi})$, the permutations $\mathbf{\Pi}, \mathbf{\Pi}\mathbf{\Pi}_{(1)}, \ldots, \mathbf{\Pi}\mathbf{\Pi}_{(B)}$ are i.i.d. uniform on $\mathbf{\Pi}_N$, hence the $B + 1$ statistics on the right-hand side are conditionally exchangeable. Therefore, conditional on $(Z, \mathbf{\Pi})$, the rank of the first coordinate among these $B + 1$ values is uniform on $\{1, \ldots, B + 1\}$ when there are no ties; with ties, the rule $T_0 > \hat{\tau}_\alpha$ is conservative. In particular,

$$\Pr\big(T_0 > \hat{\tau}_\alpha\big) \leq \alpha.$$

Thus rejecting $H_0$ when $T(Y) > \hat{\tau}_\alpha$ controls the type I error at level $\alpha$. $\qquad\square$

**A.2. Detailed Proofs of Theorem 6.2**

We present the proof of Theorem 6.2 as follows.

*Proof.* Condition on $(X^{\mathrm{tr}}, X^{\mathrm{cal}})$ (and on $\widetilde{X}^{\mathrm{cal}}$ if used). Then $\{a_f\}_{f \in \mathcal{F}}$ and $\{w_f\}_{f \in \mathcal{F}}$ are fixed, $\{w_f\}_{f \in \mathcal{F}}$ are positive, and they are independent of $(X^{\mathrm{hold}}, Y)$.

Fix any $f \in \mathcal{F}$. Let $\mu_{P,f} = \mathbb{E}_{\boldsymbol{x} \sim \mathbb{P}}[f(\boldsymbol{x})]$ and $\sigma_{P,f}^2 = \mathrm{Var}_{\boldsymbol{x} \sim \mathbb{P}}(f(\boldsymbol{x}))$ with $\boldsymbol{x} \sim \mathbb{P}$, and let $\mu_{Q,f} = \mathbb{E}_{\boldsymbol{y} \sim \mathbb{Q}}[f(\boldsymbol{y})]$ with $\boldsymbol{y} \sim \mathbb{Q}$. Since $\bar{f}$ and $\hat{\sigma}_f$ are computed from the i.i.d. calibration sample $X^{\mathrm{cal}} \sim \mathbb{P}^{n_{\mathrm{cal}}}$, by the law of large numbers,

$$\bar{f} \to \mu_{P,f} \quad \text{and} \quad \hat{\sigma}_f \to \sigma_{P,f} \qquad \text{in probability as } n_{\mathrm{cal}} \to \infty .$$

For simplicity, we take $\epsilon = 0$ in the standardization; the same argument applies when $\epsilon > 0$ by replacing $\sigma_{P,f}$ with $\sigma_{P,f} + \epsilon$.

Hence, for an independent draw $\boldsymbol{z}$ with $\mathbb{E}[f(\boldsymbol{z})^2] < \infty$,

$$a_f(\boldsymbol{z}) = \frac{f(\boldsymbol{z}) - \bar{f}}{\hat{\sigma}_f} \to \frac{f(\boldsymbol{z}) - \mu_{P,f}}{\sigma_{P,f}} \qquad \text{in probability as } n_{\mathrm{cal}} \to \infty.$$

In particular, still conditioning on $(X^{\mathrm{tr}}, X^{\mathrm{cal}})$, we have

$$\mathbb{E}_{\boldsymbol{x} \sim \mathbb{P}}[a_f(\boldsymbol{x})] \to 0 \quad \text{and} \quad \mathbb{E}_{\boldsymbol{y} \sim \mathbb{Q}}[a_f(\boldsymbol{y})] \to \Delta_f = \frac{\mu_{Q,f} - \mu_{P,f}}{\sigma_{P,f}} \qquad \text{in probability as } n_{\mathrm{cal}} \to \infty.$$

Now let $Y = \{\boldsymbol{y}_j\}_{j=1}^m \sim \mathbb{Q}^m$. By the law of large numbers and the finite second-moment assumption,

$$\frac{1}{m} \sum_{j=1}^m a_f(\boldsymbol{y}_j) \to \mathbb{E}_{\boldsymbol{y} \sim \mathbb{Q}}[a_f(\boldsymbol{y})] \qquad \text{in probability as } m \to \infty,$$

and combining with $\mathbb{E}_{\boldsymbol{y} \sim \mathbb{Q}}[a_f(\boldsymbol{y})] \to \Delta_f$ gives

$$\frac{1}{m} \sum_{j=1}^m a_f(\boldsymbol{y}_j) \to \Delta_f \qquad \text{in probability as } m \to \infty \text{ and } n_{\mathrm{cal}} \to \infty.$$

Therefore, for the weighted aggregated statistic,

$$T_{\mathrm{weight}}(Y) = \sum_{f \in \mathcal{F}} w_f \left( \frac{1}{m} \sum_{j=1}^m a_f(\boldsymbol{y}_j) \right)^2 \to C = \sum_{f \in \mathcal{F}} w_f \Delta_f^2 \qquad \text{in probability.}$$

By assumption, there exists at least one consistent $f \in \mathcal{F}$, which implies $\Delta_f \neq 0$ under $H_1 : \mathbb{P} \neq \mathbb{Q}$; since all weights are positive, this yields $C > 0$.

Next, form the pooled sample $U = X^{\mathrm{hold}} \cup Y$ with $X^{\mathrm{hold}} \sim \mathbb{P}^{n_{\mathrm{hold}}}$, and let $S$ be a uniformly random size-$m$ subset of $U$. Let $M = |S \cap Y|$ be the number of points in $S$ coming from $Y$. Then $M$ is hypergeometric and, since $n_{\mathrm{hold},m}/m \to \rho \in (0, \infty)$,

$$\frac{M}{m} \to q = \frac{m}{m + n_{\mathrm{hold}}} \to \frac{1}{1 + \rho} \in (0, 1) \qquad \text{in probability as } m \to \infty.$$

Conditional on $M$, the subset $S$ contains $M$ points from $Y$ and $m - M$ points from $X^{\mathrm{hold}}$. For each $f \in \mathcal{F}$,

$$\frac{1}{m} \sum_{\boldsymbol{z} \in S} a_f(\boldsymbol{z}) = \frac{M}{m} \cdot \frac{1}{M} \sum_{\boldsymbol{y} \in S \cap Y} a_f(\boldsymbol{y}) + \left( 1 - \frac{M}{m} \right) \cdot \frac{1}{m - M} \sum_{\boldsymbol{x} \in S \cap X^{\mathrm{hold}}} a_f(\boldsymbol{x}).$$

As $m \to \infty$, we have $M \to \infty$ and $m - M \to \infty$ in probability, so by the law of large numbers,

$$\frac{1}{M} \sum_{\boldsymbol{y} \in S \cap Y} a_f(\boldsymbol{y}) \to \mathbb{E}_{\boldsymbol{y} \sim \mathbb{Q}}[a_f(\boldsymbol{y})] \quad \text{and} \quad \frac{1}{m - M} \sum_{\boldsymbol{x} \in S \cap X^{\mathrm{hold}}} a_f(\boldsymbol{x}) \to \mathbb{E}_{\boldsymbol{x} \sim \mathbb{P}}[a_f(X)] \qquad \text{in probability.}$$

*Table 5.* Additional experiments with fixed reference size $N$ and varying query size $M$. Entries report test power as mean$_{\pm}$std (standard deviation without a leading zero). Best results in each column are bolded. $\Delta$ reports the difference between LOTTERY and the second-best method (+: outperform, −: underperform). Results are shown for Higgs ($N{=}200$) and CIFAR10-WRN8 ($N{=}100$). MMD-FUSE, MMDAgg and MMD-M cannot work when $M = 3$ in Higgs.

| | **Higgs ($N = 200$)** | | | | | | | **CIFAR10-WRN8 ($N = 100$)** | | | | | | |
| **Method** | $M = 2$ | $M = 3$ | $M = 4$ | $M = 5$ | $M = 6$ | $M = 7$ | $M = 8$ | $M = 2$ | $M = 4$ | $M = 6$ | $M = 8$ | $M = 10$ | $M = 12$ | $M = 14$ |
|---|---|---|---|---|---|---|---|---|---|---|---|---|---|---|
| MMD-FUSE | $0.392_{\pm.014}$ | $-_{\pm}-$ | $1.000_{\pm.000}$ | $1.000_{\pm.000}$ | $1.000_{\pm.000}$ | $1.000_{\pm.000}$ | $1.000_{\pm.000}$ | $0.071_{\pm.009}$ | $0.102_{\pm.013}$ | $0.149_{\pm.013}$ | $0.156_{\pm.015}$ | $0.209_{\pm.015}$ | $0.267_{\pm.010}$ | $0.322_{\pm.014}$ |
| MMDAgg | $0.607_{\pm.009}$ | $-_{\pm}-$ | $0.663_{\pm.010}$ | $0.703_{\pm.016}$ | $0.696_{\pm.014}$ | $0.774_{\pm.014}$ | $0.869_{\pm.009}$ | $0.090_{\pm.009}$ | $0.085_{\pm.011}$ | $0.127_{\pm.013}$ | $0.101_{\pm.007}$ | $0.142_{\pm.014}$ | $0.168_{\pm.009}$ | $0.204_{\pm.008}$ |
| MMD-M | $0.442_{\pm.012}$ | $-_{\pm}-$ | $0.095_{\pm.006}$ | $0.781_{\pm.010}$ | $0.921_{\pm.010}$ | $1.000_{\pm.000}$ | $1.000_{\pm.000}$ | $0.105_{\pm.010}$ | $0.161_{\pm.016}$ | $0.244_{\pm.017}$ | $0.251_{\pm.011}$ | $0.319_{\pm.017}$ | $0.417_{\pm.014}$ | $0.406_{\pm.012}$ |
| MMD-Deep | $1.000_{\pm.000}$ | $1.000_{\pm.000}$ | $1.000_{\pm.000}$ | $1.000_{\pm.000}$ | $1.000_{\pm.000}$ | $1.000_{\pm.000}$ | $1.000_{\pm.000}$ | $1.000_{\pm.000}$ | $0.142_{\pm.015}$ | $0.109_{\pm.015}$ | $0.160_{\pm.031}$ | $0.174_{\pm.031}$ | $0.269_{\pm.038}$ | $0.298_{\pm.039}$ |
| RL-TST | $1.000_{\pm.000}$ | $1.000_{\pm.000}$ | $1.000_{\pm.000}$ | $1.000_{\pm.000}$ | $1.000_{\pm.000}$ | $1.000_{\pm.000}$ | $1.000_{\pm.000}$ | $0.047_{\pm.009}$ | $0.027_{\pm.007}$ | $0.120_{\pm.024}$ | $0.149_{\pm.032}$ | $0.200_{\pm.029}$ | $0.302_{\pm.033}$ | $0.299_{\pm.038}$ |
| C2ST-L | $1.000_{\pm.000}$ | $1.000_{\pm.000}$ | $1.000_{\pm.000}$ | $1.000_{\pm.000}$ | $1.000_{\pm.000}$ | $1.000_{\pm.000}$ | $1.000_{\pm.000}$ | $0.073_{\pm.015}$ | $0.064_{\pm.012}$ | $0.184_{\pm.028}$ | $0.244_{\pm.034}$ | $0.355_{\pm.052}$ | $0.083_{\pm.035}$ | $0.087_{\pm.032}$ |
| **LOTTERY** | $1.000_{\pm.000}$ | $1.000_{\pm.000}$ | $1.000_{\pm.000}$ | $1.000_{\pm.000}$ | $1.000_{\pm.000}$ | $1.000_{\pm.000}$ | $1.000_{\pm.000}$ | $0.510_{\pm.011}$ | $\mathbf{0.687}_{\pm.016}$ | $\mathbf{0.808}_{\pm.009}$ | $\mathbf{0.836}_{\pm.011}$ | $\mathbf{0.891}_{\pm.010}$ | $\mathbf{0.929}_{\pm.007}$ | $\mathbf{0.958}_{\pm.008}$ |
| $\Delta$ (vs. 2nd best) | +0.000 | +0.000 | +0.000 | +0.000 | +0.000 | +0.000 | +0.000 | −0.490 | +0.526 | +0.564 | +0.585 | +0.536 | +0.512 | +0.552 |

Combining these limits with $\mathbb{E}_{\boldsymbol{y}\sim\mathbb{Q}}[a_f(\boldsymbol{y})] \to \Delta_f$, $\mathbb{E}_{\boldsymbol{x}\sim\mathbb{P}}[a_f(\boldsymbol{x})] \to 0$ (as $n_{\mathrm{cal}} \to \infty$), and $M/m \to q$ yields

$$\frac{1}{m} \sum_{\boldsymbol{z}\in S} a_f(\boldsymbol{z}) \to q\,\Delta_f \qquad \text{in probability as } m \to \infty \text{ and } n_{\mathrm{cal}} \to \infty,$$

and hence

$$T_{\mathrm{weight}}(S) = \sum_{f\in\mathcal{F}} w_f \left( \frac{1}{m} \sum_{\boldsymbol{z}\in S} a_f(\boldsymbol{z}) \right)^2 \to q^2 C \qquad \text{in probability.}$$

Since $q \in (0,1)$ and $C > 0$, we have $q^2 C < C$. Let $\gamma = (1 - q^2)C/4 > 0$. Then

$$\mathbb{P}\big(T_{\mathrm{weight}}(Y) \geq C - \gamma\big) \to 1 \quad \text{and} \quad \mathbb{P}\big(T_{\mathrm{weight}}(S) \leq q^2 C + \gamma\big) \to 1,$$

and for large $m$ we have $q^2 C + \gamma < C - \gamma$, so $\mathbb{P}\big(T_{\mathrm{weight}}(S) \geq T_{\mathrm{weight}}(Y)\big) \to 0$.

Now let $S_1, \ldots, S_B$ be i.i.d. uniformly sampled size-$m$ subsets of $U$ as in Section 4.4. By a union bound,

$$\mathbb{P}\Big(\max_{b\in[B]} T_{\mathrm{weight}}(S_b) < T_{\mathrm{weight}}(Y)\Big) \to 1.$$

On this event, $T_{\mathrm{weight}}(Y)$ is the largest element of the multiset $\{T_{\mathrm{weight}}(S_1), \ldots, T_{\mathrm{weight}}(S_B), T_{\mathrm{weight}}(Y)\}$. Since $1/(B + 1) < \alpha$, the empirical $(1 - \alpha)$-quantile $\hat{\tau}_\alpha$ is strictly below the maximum, hence $T_{\mathrm{weight}}(Y) > \hat{\tau}_\alpha$. Therefore,

$$\mathbb{P}\big(T_{\mathrm{weight}}(Y) > \hat{\tau}_\alpha\big) \to 1 \qquad \text{as } m \to \infty \text{ and } n_{\mathrm{cal}} \to \infty,$$

which proves consistency. $\qquad\qquad\qquad\qquad\qquad\qquad\qquad\qquad\qquad\qquad\qquad\qquad\qquad\qquad\square$

## B. Experimental Details

### B.1. Extra Experimental Results

This appendix reports additional experimental results (Table 5 and Table 6) that complement the main paper and further assess the robustness of LOTTERY under severe sample-size asymmetry. Across all supplementary benchmarks, LOTTERY consistently achieves strong test power in few-query regimes and scales favorably as either the reference size $N$ or the query size $M$ increases.

When $M$ is fixed and $N$ grows, LOTTERY rapidly approaches near-perfect power on vision benchmarks, while kernel-based baselines improve more slowly. Conversely, when $N$ is fixed and $M$ varies, LOTTERY exhibits the largest gains in the most data-scarce settings and converges to the performance of strong learning-based baselines as $M$ becomes moderately large. On Higgs, many learning-based methods saturate to power $1.0$, whereas on CIFAR10-WRN28 variants, LOTTERY provides clear advantages in high-dimensional, few-shot settings.

These results further confirm that reference-only learning with uncertainty-guided aggregation is particularly effective under extreme imbalance.

*Table 6.* Additional experiments with fixed query size $M$ and varying reference size $N$. Entries report test power as mean$_\pm$std (standard deviation without a leading zero). Results are shown for Higgs with $M = 2$ and CIFAR10-WRN28 with $M = 4$.

| Method | Higgs ($M = 2$) | | | | | | | CIFAR10-WRN28 ($M = 4$) | | | | | | |
|---|---|---|---|---|---|---|---|---|---|---|---|---|---|---|
| | $N = 50$ | $N = 80$ | $N = 100$ | $N = 120$ | $N = 150$ | $N = 180$ | $N = 200$ | $N = 40$ | $N = 50$ | $N = 60$ | $N = 70$ | $N = 80$ | $N = 90$ | $N = 100$ |
| MMD-FUSE | $0.302_{\pm.011}$ | $0.294_{\pm.013}$ | $0.386_{\pm.013}$ | $0.415_{\pm.010}$ | $0.370_{\pm.017}$ | $0.387_{\pm.021}$ | $0.413_{\pm.012}$ | $0.091_{\pm.010}$ | $0.112_{\pm.012}$ | $0.097_{\pm.009}$ | $0.115_{\pm.011}$ | $0.102_{\pm.009}$ | $0.101_{\pm.009}$ | $0.106_{\pm.007}$ |
| C2ST-L | $1.000_{\pm.000}$ | $1.000_{\pm.000}$ | $1.000_{\pm.000}$ | $1.000_{\pm.000}$ | $1.000_{\pm.000}$ | $1.000_{\pm.000}$ | $1.000_{\pm.000}$ | $0.044_{\pm.012}$ | $0.058_{\pm.013}$ | $0.085_{\pm.014}$ | $0.061_{\pm.014}$ | $0.105_{\pm.013}$ | $0.135_{\pm.018}$ | $0.052_{\pm.012}$ |
| MMD-Agg | $0.399_{\pm.019}$ | $0.518_{\pm.021}$ | $0.625_{\pm.014}$ | $0.594_{\pm.013}$ | $0.611_{\pm.009}$ | $0.580_{\pm.016}$ | $0.615_{\pm.008}$ | $0.083_{\pm.010}$ | $0.122_{\pm.013}$ | $0.089_{\pm.009}$ | $0.090_{\pm.008}$ | $0.093_{\pm.011}$ | $0.089_{\pm.009}$ | $0.080_{\pm.009}$ |
| MMD-M | $0.039_{\pm.005}$ | $0.159_{\pm.016}$ | $0.295_{\pm.014}$ | $0.442_{\pm.014}$ | $0.314_{\pm.015}$ | $0.436_{\pm.011}$ | $0.453_{\pm.016}$ | $0.148_{\pm.010}$ | $0.183_{\pm.012}$ | $0.160_{\pm.008}$ | $0.191_{\pm.014}$ | $0.179_{\pm.010}$ | $0.171_{\pm.010}$ | $0.163_{\pm.008}$ |
| RL-TST | $1.000_{\pm.000}$ | $1.000_{\pm.000}$ | $1.000_{\pm.000}$ | $1.000_{\pm.000}$ | $1.000_{\pm.000}$ | $1.000_{\pm.000}$ | $1.000_{\pm.000}$ | $0.042_{\pm.009}$ | $0.030_{\pm.009}$ | $0.056_{\pm.011}$ | $0.045_{\pm.014}$ | $0.055_{\pm.016}$ | $0.084_{\pm.021}$ | $0.031_{\pm.012}$ |
| MMD-Deep | $1.000_{\pm.000}$ | $1.000_{\pm.000}$ | $1.000_{\pm.000}$ | $1.000_{\pm.000}$ | $1.000_{\pm.000}$ | $1.000_{\pm.000}$ | $1.000_{\pm.000}$ | $0.105_{\pm.013}$ | $0.123_{\pm.027}$ | $0.107_{\pm.012}$ | $0.127_{\pm.017}$ | $0.117_{\pm.016}$ | $0.109_{\pm.015}$ | $0.144_{\pm.022}$ |
| **LOTTERY** | $1.000_{\pm.000}$ | $1.000_{\pm.000}$ | $1.000_{\pm.000}$ | $1.000_{\pm.000}$ | $1.000_{\pm.000}$ | $1.000_{\pm.000}$ | $1.000_{\pm.000}$ | $0.517_{\pm.018}$ | $0.525_{\pm.009}$ | $0.647_{\pm.007}$ | $0.646_{\pm.017}$ | $0.656_{\pm.014}$ | $0.661_{\pm.011}$ | $0.669_{\pm.023}$ |

*Table 7.* Complementarity of individual RDR families. We report test power at $\alpha = 0.05$ over 10 independent trials with $N = 4000$ reference samples and 200 permutations per test. Bold numbers indicate the best individual RDR in each row. Bold LOTTERY indicates that the aggregated test exceeds all individual RDRs.

| Alternative | $m$ | LOF | kNN | Mahalanobis | LOTTERY |
|---|---|---|---|---|---|
| mean_shift | 20 | .197 | .192 | **.221** | **.265** |
| | 50 | .278 | .288 | **.320** | **.442** |
| | 100 | .563 | .573 | **.629** | **.661** |
| | 200 | .717 | .765 | **.773** | **.871** |
| variance_scale | 20 | .196 | .197 | **.217** | **.268** |
| | 50 | .301 | .299 | **.338** | **.444** |
| | 100 | .549 | .569 | **.607** | **.656** |
| | 200 | .731 | .735 | **.779** | **.865** |
| skew_variance | 20 | **.313** | .275 | .295 | **.362** |
| | 50 | **.447** | .403 | .444 | **.536** |
| | 100 | **.727** | .689 | .720 | **.760** |
| | 200 | **.860** | .824 | .853 | **.911** |
| point_contam | 20 | .121 | .103 | **.122** | **.167** |
| | 50 | .203 | .219 | **.270** | **.317** |
| | 100 | .379 | .416 | **.547** | **.563** |
| | 200 | .525 | .621 | **.793** | **.825** |

## B.2. Complementarity of Individual RDR Families

To further examine the motivation for using multiple RDR families, we construct four synthetic alternatives with the same reference distribution $P = \mathcal{N}(0, I_d)$, where $d = 10$. Unless otherwise stated, the perturbation strength is set to $s = 2.0$. The alternatives are designed to emphasize different types of distributional changes. In the mean_shift setting, the query distribution is $Q = \mathcal{N}(\delta e_1, I_d)$ with $\delta = 1.0$, targeting global location changes. In the variance_scale setting, $Q = \mathcal{N}(0, (1 + \epsilon)I_d)$ with $\epsilon = 0.10$, targeting global scale changes. In the skew_variance setting, $Q = \mathcal{N}(0, \text{diag}(s_1, \ldots, s_d))$, where the first three coordinates have variance 2.0, the next three have variance 0.4, and the remaining coordinates have variance 1.0, producing heterogeneous coordinate-wise changes. In the point_contam setting, $Q = (1 - \epsilon)\mathcal{N}(0, I_d) + \epsilon\mathcal{N}(4e_1, 0.01I_d)$ with $\epsilon = 0.15$, creating a small cluster of localized outliers.

Table 7 reports the test power of each individual RDR and LOTTERY across query sizes $m \in \{20, 50, 100, 200\}$. The best individual RDR varies across alternatives and query sizes, confirming that the RDR families capture non-redundant signals. LOTTERY consistently outperforms every individual RDR by aggregating these complementary signals.

## B.3. Implementation Details

All experiments were conducted on a remote Linux server equipped with an NVIDIA A100 GPU, an 8-core CPU, and 16 GB of memory. All methods were implemented in **Python** using the **PyTorch** framework. Kernel-based baselines additionally rely on NumPy and SciPy for distance computations and U-statistics.

Random seeds are fixed across runs, and all results are averaged over multiple independent trials. The complete implementation, including data generation, training pipelines, and evaluation scripts, is provided in the supplementary materials.

## B.4. Details of Baselines

We compare LOTTERY with representative nonparametric two-sample testing baselines, including both kernel-based and learning-based methods.

**MMD-M**   (Garreau et al., 2017) Maximum Mean Discrepancy with a single Gaussian kernel, using the median heuristic for bandwidth selection.
**Code:** https://github.com/yeager20001118/AdapTesting

**MMD-FUSE**   (Biggs et al., 2023) A multi-kernel MMD test that adaptively fuses a family of kernels while preserving finite-sample type I error control without data splitting.
**Code:** https://github.com/antoninschrab/mmdfuse

**MMDAgg**   (Schrab et al., 2023) Aggregates multiple MMD statistics via a calibrated aggregation rule to improve robustness across kernel choices.
**Code:** https://github.com/antoninschrab/mmdagg

**C2ST-L**   (Lopez-Paz & Oquab, 2017; Kim et al., 2021) Classifier two-sample test based on training a binary classifier and using its logits as the test statistic.
**Code:** https://github.com/yeager20001118/AdapTesting

**MMD-Deep**   (Liu et al., 2020) Deep-kernel MMD with representation learning on a training split and testing on a held-out set.
**Code:** https://github.com/fengliu90/DK-for-TST

**RL-TST**   (Tian et al., 2025) A reinforcement-learning-based two-sample test that optimizes representations to maximize test power under a train–test split.
**Code:** https://github.com/yeager20001118/A-Unified-Data-Representation-Learning-for-Non-paramet

Unless otherwise specified, all baselines are tuned following their original papers.

## B.5. Details of Datasets Generation

**BLOB (synthetic).** We follow the BLOB data-generation protocol used in prior two-sample testing work (Gretton et al., 2012; Liu et al., 2020). In brief, each sample is drawn from a mixture of Gaussians arranged on a grid in $\mathbb{R}^2$ (the reference distribution), while the query distribution is obtained by perturbing the mixture structure (e.g., by shifting a subset of components and/or modifying their covariances). This benchmark is designed to stress tests that must detect local and non-Gaussian distributional changes.

**CIFAR-10 adversarial detection.** We treat clean CIFAR-10 images as reference samples and adversarially perturbed images as query samples. Given an input image $x$ and label $y$, projected gradient descent (PGD) constructs an adversarial example by iteratively taking gradient steps to increase the classification loss and then projecting back to a constraint set (here an $\ell_\infty$ ball of radius $\epsilon$ around $x$ and the valid pixel range). Concretely,

$$x^{(t+1)} = \Pi_{\mathcal{B}_\infty(x,\epsilon)}\big(x^{(t)} + \eta \operatorname{sign}(\nabla_x \mathcal{L}(\theta, x^{(t)}, y))\big),$$

where $\Pi$ denotes projection and $\eta$ is the step size.

**Embedding extractors (ResNet-18 / WRN-28).** For image-based experiments, we embed each image using a fixed, pretrained classifier and apply all two-sample tests in the resulting feature space. We consider (i) ResNet-18 (He et al., 2016), which is a residual network composed of four stages of basic residual blocks (commonly configured as $[2, 2, 2, 2]$ blocks), and (ii) WideResNet-28 (Zagoruyko & Komodakis, 2016), a widened residual architecture with depth 28. Unless stated otherwise, we use the penultimate-layer (global-average-pooled) features as embeddings.

