# OpenReview forum: "LOTTERY: Learning from Reference-Only Samples in Two-Sample Testing under Size Asymmetry"
_ICML.cc/2026/Conference — ICML 2026 regular_

### Official Review · Reviewer_bzXy · 2026-03-09

**Soundness:** 4
**Presentation:** 4
**Significance:** 3
**Originality:** 3
**Overall Recommendation:** 4
**Confidence:** 3

**Summary:**

This paper studies two-sample testing under sample-size asymmetry, where reference data are abundant but query data are scarce. Existing learning-based methods become unstable in both representation learning and hypothesis testing in the extreme few-shot query regime. To address this issue, the authors propose a shift in perspective: representations are learned solely from the reference data, while the entire query set is used exclusively for testing. Concretely, the paper introduces the LOTTERY framework, which learns four complementary Reference-Dependent Representations (RDRs) from the reference distribution, combines them using weights that balance stability and sensitivity, and performs inference via a pooled permutation test. The authors provide theoretical guarantees for finite-sample type I error control and consistency, and demonstrate empirical improvements over existing baselines in the extreme few-shot query regime.

**Compliance With Llm Reviewing Policy:**

Affirmed.

**Final Justification:**

The paper proposes a principled testing framework for benchmark selection using randomized discrepancy ranks, with level-$\alpha$ type I error control that clearly distinguishes it from one-class classification approaches. The theoretical development is sound and the experimental evaluation is adequate.

The rebuttal fully addressed my concerns. The distinction between OCC (no FPR control) and LOTTERY (level-α test) clarifies the contribution, and the explanation of Theorem 6.2 as a standard asymptotic sanity check is reasonable. That said, the conceptual distance from existing anomaly detection and OCC methods remains moderate, as the individual components (rank-based discrepancy scores, their combination) draw on established techniques. I maintain my score of 4.

**Key Questions For Authors:**

Please see the weaknesses section.

**Limitations:**

Partially. The Impact Statement briefly mentions that misuse may arise without consideration of assumptions or domain context, but the paper does not include a dedicated Limitations section. Important limitations such as the absence the reliance on a fixed set of four predefined RDR families and the data efficiency loss from the three-way reference split are not discussed.

**Strengths And Weaknesses:**

### Strengths
- Two-sample testing with few-shot queries is a practically relevant setting, yet existing learning-based methods often implicitly assume more balanced sample sizes. The motivation for addressing this problem directly is therefore clear and compelling.
- The empirical gains in the few-shot regime are large and consistent across experiments.
- The ablation study clearly isolates the contribution of uncertainty-guided selection. In particular, the fact that the largest improvements occur when the number of query samples is smallest supports the claim that the weighting mechanism is addressing the main bottleneck of the problem.
- The theoretical analysis provides guarantees for the validity of the proposed methodology.

### Weaknesses
- The conceptual novelty appears somewhat limited. The method largely follows the standard paradigm of one-class classification (OCC), in which a model is trained only on normal/reference data and new samples are evaluated against it. The individual RDRs are already well-known anomaly detection scores, and combining multiple detectors is also standard practice in anomaly detection. Although the paper applies these ideas to the new setting of two-sample testing, the intellectual gap between OCC and LOTTERY seems relatively small.
- Theorem 6.2 does not appear to reflect the regime for which LOTTERY is primarily motivated. While the method is designed for settings with very limited query samples, the consistency result requires m to infinity. Although such an asymptotic result is conventional, its relevance to the paper's main motivation seems somewhat limited.

### Minor
- The paper does not include a dedicated Related Work section. Instead, prior work is dispersed across the introduction and preliminaries, which makes it harder to clearly assess the paper's positioning and the precise scope of its contribution.

---

> ### Author Rebuttal · Authors · 2026-03-30
>
> Dear Reviewer bzXy,
>
> We sincerely thank you for your insightful comments and suggestions, and we will clarify your questions point by point below:
>
> > Response to Weaknesses:
>
> **R1 for W1**: We would like to firstly clarify the conceptual contribution of our work, compared to the OCC framework. The existing OCC's detectors may use similar scores but offer no control over false positive rates. However, LOTTERY transforms a family of such scores into **a level-$\alpha$ two-sample test with finite-sample type I error control**. In the following, we want to state our contributions from both perspectives of conceptual novelty and technical novelty. Conceptually, LOTTERY is **the first to learn representations for two-sample testing under extreme class imbalance**, opening a concrete direction for future score design in the field of two-sample testing. From the perspective of technical contribution, we propose the powerful **uncertainty-guided weighted selection** which is a novel machanism to aggregate different RDRs that target different aspects of distributional discrepancy, *as verified in the ablation study in Section 7.3 and agreed by the reviewers aeK3 and EKX2*. Thanks for pointing out that concerns and we will also emphasize these contributions more clearly in the revised manuscript to make the key points easier for readers to grasp.
>
> **R2 for W2**: Thank you for this observation. We would like to clarify the functionality of Theorem 6.2. Consistency is a standard asymptotic sanity check in the two-sample testing scenario. Its purpose is to rule out degenerate behavior, so it guarantees that the test is not trivially powerless, i.e., as data grows, the test will eventually detect any true difference. Without this result, readers may reasonably ask whether the aggregated test might be too conservative to control type I error trivially (by never rejecting). Consistency confirms this is not the case. Moreover, LOTTERY aggregates multiple OCC-based scores, and OCC classifiers by design do not account for the sample correlation structure inherent in two-sample testing. Theorem 6.2 ensures that despite this aggregation, LOTTERY remains a method of two-sample test with guaranteed consistency, which is a property OCC classifiers alone do not possess. We agree the finite-sample regime is the primary motivation, and we will add a clarifying remark to better position Theorem 6.2 in the revision.
>
> **R3 for W3**: Thank you for this suggestion. We agree that a dedicated Related Work section would improve clarity and make the positioning of our contribution easier to assess. In the revised manuscript, we will consolidate the currently scattered discussion into a dedicated Related Work section covering: (1) kernel-based two-sample testing and aggregation methods (MMDAgg, MMD-FUSE), (2) learning-based two-sample tests (C2ST, MMD-Deep, RL-TST), and (3) sequential two-sample testing approaches (as suggested by Reviewer aeK3). This will more clearly present how LOTTERY differs from and builds upon each line of work.
>
> > Response for Limitations:
>
> **R for L**: Thank you for this important feedback. We will add a dedicated Limitations section in the revised manuscript, explicitly discussing the following points:
>
> 1) The current framework uses Mahalanobis, $k$NN, LOF, and ME-RDR. While these cover a range of global-to-local distributional aspects and include a theoretically consistent component (ME-RDR, Remark 4.2), they are not guaranteed to be optimal for all application domains. The framework is scalable for additional RDR families (e.g., deep one-class representations) can be incorporated, but we have not explored automated RDR selection.
>
> 2) LOTTERY is designed for settings with abundant reference data and scarce queries. When $N$ is small, the three-way split leaves insufficient data for learning informative RDRs, and kernel baselines that use all data directly may perform better.
>
> Best regards,
>
> Authors of Submission 13176

---

> > ### Author Rebuttal · Reviewer_bzXy · 2026-04-03
> >
> > The authors adequately addressed my two concerns. The distinction between OCC and LOTTERY clarifies the contribution, and the explanation of Theorem 6.2 as a standard asymptotic sanity check is reasonable. I maintain my positive score.

---

> > > ### Author Response · Authors · 2026-04-03
> > >
> > > Thank you very much for your positive update and for carefully considering our rebuttal. We are very glad that the distinction between OCC and LOTTERY is now clearer, and we sincerely appreciate your recognition of the motivation and contribution of our work in this area. We also thank you for your positive view of our explanation of Theorem 6.2. Our paper will definitely be improved by merging your thoughts and suggestions!
> > >
> > > Since your concerns have now been fully addressed, as suggested by the acknowledgement option (a): *If you select this option, please consider adjusting your score accordingly*, we would be very grateful if your final score could be calibrated accordingly.
> > >
> > > Thank you again for your thoughtful and constructive feedback throughout the review process.

---

### Official Review · Reviewer_EKX2 · 2026-03-11

**Soundness:** 4
**Presentation:** 4
**Significance:** 3
**Originality:** 2
**Overall Recommendation:** 4
**Confidence:** 3

**Summary:**

The paper proposes a learning-based two sample test for the setting where only data from one distribution is available. The proposed method employs several existing test statistic-like metrics and aggregates in a statistical test that demonstrates good performance on a few small batch out-of-support detection tasks.

**Compliance With Llm Reviewing Policy:**

Affirmed.

**Key Questions For Authors:**

- the problem of learning-based two-sample testing is reminiscent of semi-supervised OOD detection [1, 2, 3] and PU learning [4, 5]. In both cases, in addition to the reference data, an unlabeled set of samples from both distributions is used in the representation learning stage to improve the power of the test. It would be interesting to understand whether approaches proposed for these problems are applicable in the setting of the current paper and, if so, how the proposed method compares to them.
- how has the set of RDRs been chosen? While the motivation for covering both representation that capture local and global features is clearly stated, it is not explained why these specific RDRs (i.e. the ones in section 4.2) were chosen for this purpose.

[1] - https://www.jmlr.org/papers/volume11/blanchard10a/blanchard10a.pdf

[2] - https://arxiv.org/abs/1908.04951

[3] - https://arxiv.org/pdf/2012.05825

[4] - https://cseweb.ucsd.edu/~elkan/posonly.pdf

[5] - https://arxiv.org/abs/1703.00593

**Limitations:**

The paper discusses some of the limitations of the proposed method, including the poorer performance in the low sample regime.

**Strengths And Weaknesses:**

**Strengths:**

- extensive experimental ablations that also highlight when the proposed method breaks (e.g. low sample regime) as well as where its strengths are (e.g. small query size regime)
- the setting and method are clearly described, and the mathematical derivations are correct

**Weaknesses:**

- the proposed method simply combines a series of existing metrics, already employed in prior works on out-of-distribution detection and/or two-sample tests. Having said that, the manner in which these are combined is, while simple, novel, to the best of my knowledge.
- while the goal of the proposed method is to test validity of the null $\mathbb{P} = \mathbb{Q}$, in reality, the experiments section focuses on a narrower, and hence easier, version of this test. Namely, all the experiments focus on settings where the null hypothesis can be written as $supp(\mathbb{P}) = supp(\mathbb{Q})$, where $supp$ denotes the support of a distribution. Since the out-of-support detection problem is easier than deciding if $\mathbb{P} = \mathbb{Q}$, it is conceivable that only 4 query samples are sufficient for good performance as illustrated in Table 1. However, the problem becomes significantly more challenging if $\mathbb{P}$ and $\mathbb{Q}$ are different but share the same support.
- the experimental analysis is carried out on a rather limited set of only three datasets, two of which are quite synthetic in nature. It would help to make the results more convincing if experiments on more datasets were included in the manuscript, similar to other works on out-of-distribution detection [1, 2], PU learning [3, 4] etc.

[1] - https://arxiv.org/pdf/2012.05825

[2] - https://arxiv.org/abs/2106.03004

[3] - https://cseweb.ucsd.edu/~elkan/posonly.pdf

[4] - https://arxiv.org/abs/1703.00593

---

> ### Author Rebuttal · Authors · 2026-03-30
>
> Dear Reviewer EKX2,
>
> Thank you for your careful reading and valuable feedback. We will clarify your questions point by point below:
>
> > Response to Weaknesses:
>
> **R1 for W1**: Thank you for the positive comment on our technical novelty of weighted selection machanism, which is not complex but effective as shown in the ablation study in Section 7.3. Furthermore, we would like to clarify that the contribution of LOTTERY extends beyond combining existing scores: LOTTERY is the first to learn representations for two-sample testing under extreme class imbalance, opening a concrete direction for future score design in the field of two-sample testing. We will point out that in our revised manuscript so that the takeaway is clearer to readers.
>
> **R2 for W2**: Thank you for pointing out this distinction. We would like to clarify that our experiments do cover the harder same-support setting. BLOB explicitly satisfies $\text{supp}(\mathbb{P}) = \text{supp}(\mathbb{Q})$ with only covariance differences, and HIGGS involves a pure density shift with full support overlap. For CIFAR10-WRN28, the t-SNE visualization (https://anonymous.4open.science/r/ICML2026_LOTTERY_Rebuttal-DC6F/visualization_dataset.pdf) shows substantially more overlap between clean and adversarial embeddings compared to ResNet18, confirming it is a harder near-same-support case. Despite this, LOTTERY achieves test power of 0.517–0.669 across $N=40$ to $N=100$ while others' power of around 0.1 ($M=4$, Table 5), demonstrating that our method remains effective even without a support gap. We will explicitly highlight which settings satisfy $\text{supp}(\mathbb{P}) = \text{supp}(\mathbb{Q})$ in the revised manuscript.
>
> **R3 for W3**: Thank you for the helpful suggestion. We agree that evaluating on more datasets would strengthen the paper. We have run additional experiments on a real-world text task: Olympics 2024 News vs. Gemini-Pro generated news (Machine-Generated Text detection). LOTTERY outperforms all baselines at low $m$ and matches learning-based methods at high $m$ without requiring extra adapter training (results provided in https://anonymous.4open.science/r/ICML2026_LOTTERY_Rebuttal-DC6F/Olympics_MGT_results.md). We are also running CIFAR100 adversarial detection experiments (3/4 of the similar work running that) and will include all these results in the revised manuscript.
>
> > Answer to Key Questions:
>
> **A1 for Q1**: Thank you for drawing this connection. We appreciate the opportunity to clarify the relationship between our setting and semi-supervised OOD detection / PU learning.
>
> The referenced semi-supervised OOD methods and PU learning approaches share a critical assumption that an **unlabeled mixture of samples from both distributions** is available during the training. Importantly, OOD / PU learning is a complementary setting of our problem setting. They are powerful when unlabeled mixed data is not only available but also requires sufficient out-of-distribution representation in the training pool, as explored in a two-sample testing work setting [1]. And our proposed LOTTERY are designed for setting where query data is too scarce for any learning, and all representation power must come from the reference distribution alone.
>
> [1]. A Unified Data Representation Learning for Non-parametric Two-sample Testing, UAI 2025.
>
> **A2 for Q2**: Thank you for this helpful question. The RDRs were chosen to provide **complementary views of distributional change**. Practical discrepancies may appear as global mean/covariance shifts, local geometric or support-related deviations, or localized density irregularities. Accordingly, **Mahalanobis-RDR** targets global location/covariance changes, **kNN-RDR** captures local neighborhood geometry and support mismatch, and **LOF-RDR** emphasizes local relative-density anomalies. For example, an abnormal sample may lie near the boundary of the reference population and thus be hard to detect through global summaries, while a small anomalous cluster may be primarily visible through local density. We will revise the paper to make this selection principle more explicit and to emphasize that the framework is modular and can be extended with additional RDRs when needed. Moreover, we conduct additional experiments under carefully designed synthetic scenarios to illustrate the complementary signals captured by different RDRs, as discussed in our response [R2] to Reviewer aeK3.
>
> Best regards,
>
>  Authors of Submission 13176

---

> > ### Author Rebuttal · Reviewer_EKX2 · 2026-04-03
> >
> > I would like to thank the authors for the rebuttal and for addressing my comments. I maintain my positive score for the paper.

---

> > > ### Author Response · Authors · 2026-04-06
> > >
> > > Thank you very much for your positive update and for carefully considering our rebuttal. We are very glad that the technical novelty concern has been clarified by our responses and that our additional experiments further confirm the effectiveness of our method. Our paper will definitely be improved by merging your thoughts and suggestions!
> > >
> > > Since your concerns have now been fully addressed, as suggested by the acknowledgement option (a): *"If you select this option, please consider adjusting your score accordingly"*, we would be very grateful if your final score could be calibrated accordingly.
> > >
> > > Thank you again for your thoughtful and constructive feedback throughout the review process.

---

### Official Review · Reviewer_aeK3 · 2026-03-12

**Soundness:** 3
**Presentation:** 3
**Significance:** 3
**Originality:** 3
**Overall Recommendation:** 4
**Confidence:** 4

**Summary:**

This paper addresses the two-sample testing problem in scenarios where the sample sizes of the two groups are highly imbalanced. The work treats the larger sample as a reference set and proposes the concept of reference-dependent representation (RDR) to evaluate the similarity of a data point to the reference distribution. The authors construct three heuristic RDRs that capture local similarity patterns and the global structure of the reference sample distribution. The aggregated RDR computed from the smaller sample is then used as a test statistic and evaluated against the union of the two samples. The type-I error is controlled through a permutation procedure, and the testing power is further improved using an RDR-weighted strategy.

**Compliance With Llm Reviewing Policy:**

Affirmed.

**Key Questions For Authors:**

1. The paper provides an ablation study to evaluate the importance of the RDR-weighted strategy. However, does it also evaluate the individual contributions of the three RDR variants to the observed improvement in testing power? The selection of these RDRs appears somewhat heuristic, so a more systematic analysis of their respective roles would strengthen the paper.

2. Another way to formulate this problem is through the joint distribution $p(x, y)$, where $x$ denotes the sample features and $y$ denotes the sample membership indicator. Under this view, a two-sample test is equivalent to testing whether $p(x, y) = p(x)p(y)$. In this formulation, class imbalance is incorporated naturally. There are prior works (see below) that estimate the mutual information between $X$ and $Y$ and do not require a train/test split. I think these works would be valuable to discuss in the related work section. In particular, can the authors comment on the advantages of their proposed strategy relative to these approaches?

3.  The presented RDRs seems to be largely heuristic. is it possible to construct a mathematical framework for seeking RDR?

Lhéritier, Alix, and Frédéric Cazals. "A sequential non-parametric multivariate two-sample test." *IEEE Transactions on Information Theory* 64, no. 5 (2018): 3361--3370.

Li, W., Kadambi, P., Saidi, P., Ramamurthy, K. N., Dasarathy, G., & Berisha, V. "Active Sequential Two-Sample Testing." *Transactions on Machine Learning Research*.

**Limitations:**

It seems there's not discussion on the limitation.

**Strengths And Weaknesses:**

**Soundness**
Overall, the work is sound. Conventional two-sample tests evaluate the discrepancy between two samples under the assumption that the sample sizes are comparable. However, in scenarios with extreme class imbalance, the main bottleneck lies in the limited size of the smaller sample. To address this, the work proposes embedding the distributional representation of the large sample through the reference-dependent representation (RDR), thereby leveraging the statistical strength of the larger dataset. Meanwhile, the smaller sample is fully utilized when computing the test statistic.

**Presentation**
The paper could benefit from clearer explanations of several concepts. In particular, the meanings of “ME” in ME-RDR and “LOF” in LOF-RDR should be explicitly clarified. In addition, the presentation could be improved by expanding the motivation for introducing multiple RDRs and how the three RDR variants complement one another. The description of the permutation procedure in Section 4.4 is also somewhat imprecise. Strictly speaking, a permutation test relies on exhaustively shuffling sample memberships  rather than uniform resampling to construct the null distribution.

**Significance**
The work is significant as it addresses a practical and relatively underexplored setting for two-sample testing, namely cases with severe sample size imbalance.

**Originality**
The work demonstrates originality in two aspects: (1) the considered problem setting—two-sample testing under extreme sample imbalance—is relatively new, and (2) the proposed strategy leverages the large sample to construct a metric that evaluate the distance from the reference distribution, and, tests the similarity between the small sample and the reference distribution.

---

> ### Author Rebuttal · Authors · 2026-03-30
>
> Dear Reviewer aeK3,
>
> Thank you for your thoughtful comments and constructive suggestions. We will clarify your questions point by point below:
>
> > Response to Weaknesses:
>
> > W1 (Presentation): the meanings of “ME” in ME-RDR and “LOF” in LOF-RDR should be explicitly clarified ...
>
> **R1**: Thanks for pointing out that ambiguity. "ME" stands for *Mean Embedding*, referring to the kernel mean embedding representation, "LOF" stands for *Local Outlier Factor*, a density-based anomaly score [1]. We will spell out both acronyms at first use in the revised manuscript.
>
> > W2 (Presentation): improved by expanding the motivation ...
>
> **R2**: Thanks for the advice! We will expand both motivation below and in the revised manuscript.
>
> 1). The reason is that there is no single RDR is uniformly powerful against all scenarios of distribution discrepancy detection, since it may appear as global mean/covariance shifts, local geometric deformations, support mismatch or localized density irregularities. Applying multiple RDRs aim to detect even multiple combined discrepancies.
>
> 2). To empirically validate the complementarity of these 3 RDR families, we construct four synthetic scenarios on $\mathbb{R}^{10}$ following the High Dimension Gaussian Mixture (HDGM) framework [2], which is displayed at https://anonymous.4open.science/r/ICML2026_LOTTERY_Rebuttal-DC6F/RDR_individual.md. The results confirm that the three RDR variants can capture non-redundant distributional scales. Across all 4 alternatives, LOTT exceeds every individual RDR on all 4 settings by aggregating these complementary signals.
>
> We will add the detail of this experiment and analysis of motivation in the revised manuscript.
>
> > W3 (Presentation): permutation procedure in Section 4.4 is also somewhat imprecise ...
>
> **R3**: We thank the reviewer for the clarification. Our method indeed does not exhaustively enumerate all $\binom{n_{\mathrm{hold}}+m}{m}$ label assignments. Rather, it uses a **Monte Carlo permutation/randomization test** [3]: after pooling $X^{\mathrm{hold}}$ and $Y$, we repeatedly sample a size-$m$ subset uniformly at random, equivalently permuting group labels while preserving sample sizes. We will revise the manuscript to state this more precisely.
>
> > Answer to Key Questions:
>
> **A1 for Q1**: Thanks! We agree that a more systematic analysis of how the individual RDR contributes to the power improvement would strengthen the paper. Our additional experiments for reviewer EKX2 suggest why such analysis is important. In one case, Mahalanobis-RDR attains a very large raw score (around 70) but receives a small weight (0.03) due to large variance indicating that this RDR is unstable, and its single power is only about 0.3. By contrast, LOF-RDR has a much smaller raw score (around 1.13) but a much larger weight (3.75) reflecting much greater stability and utility, and its single power is about 1.0. Without the proposed weighting, the large but unstable Mahalanobis-RDR would dominate the aggregation and reduce the final power to around 0.3; with weighting, the power increases to about 0.99. We will incorporate this type of example more clearly in the revision for better understanding the role of selection.
>
> **A2 for Q2**: Thanks for this helpful suggestion. We agree that two-sample testing can be equivalently formulated as testing independence, and we will add this perspective and the suggested references in the related work.
>
> We note that class imbalance poses the same challenge under this formulation, e.g., when $p(y=0)=0.99$, $p(y=1)=0.01$, MI estimators also struggle in this imbalanced setting. Our key advantage is that we learn various RDRs that can be used to replace raw $x$, which provides a better-conditioned input and can potentially extend to help this scenario. Additionally, our method only requires a split on the reference distribution and is zero-shot with respect to the test distribution, which is a practically important distinction in deployment settings.
>
> **A3 for Q3**: We acknowledge this is a valid concern. A fully unified mathematical framework for seeking an optimized RDR remains an open problem in the field,  as no single score metric can fully capture all signals of a distribution. However, our framework is grounded in Definition 4.1, which provides a principled criterion: any statistic qualifying as an RDR can be incorporated, and LOTTERY will leverage it to enhance test power. A natural direction for future work is to formulate RDR construction as a power maximization problem, which we will discuss explicitly in the revision.
>
> **Regarding the limitations**, we will add a section to discuss that in revised manuscript.
>
> [1]. LOF: identifying density-based local outliers, SIGMOD 2000.
>
> [2]. Learning Deep Kernels for Non-Parametric Two-Sample Tests, ICML 2020.
>
> [3]. Permutation P-values Should Never Be Zero, Statistical Applications in Genetics and Molecular Biology 2010.
>
> Best regards,
>
> Authors of Submission 13176

---

> > ### Author Rebuttal · Reviewer_aeK3 · 2026-04-03
> >
> > Thank you for your response. I will keep my positive score.

---

> > > ### Author Response · Authors · 2026-04-06
> > >
> > > Thank you very much for your positive update and for carefully considering our rebuttal. We are very glad that concerns regarding the selection of RDRs have been clarified by our responses and that our additional experiments further confirm the effectiveness of our method. Our paper will definitely be improved by merging your thoughts and suggestions!
> > >
> > > Since your concerns have now been fully addressed, as suggested by the acknowledgement option (a): "If you select this option, please consider adjusting your score accordingly", we would be very grateful if your final score could be calibrated accordingly.
> > >
> > > Thank you again for your thoughtful and constructive feedback throughout the review process.

---

### Decision · Program_Chairs · 2026-04-30

**Decision:**

Accept (regular)

**Comment:**

Reviewers agreed that the paper addresses an important and underexplored setting of two-sample testing, and found the proposed  framework to be technically sound with clear gains in the few-shot regime. Theoretical guarantees and ablations support the approach, making this a solid contribution.

For the final version, the authors are encouraged to take the great discussions here into account. Consider clearer positioning relative to one-class classification and other related work. Also consider better articulation of conceptual novelty and the role of various design choices. This will further strengthen the paper.